# Molecular and structural basis of oligopeptide recognition by the Ami transporter system in pneumococci

Martín Alcorlo[1], Mohammed R. Abdullah[2], Leif Steil[3], Francisco Sotomayor[1], Laura López-de Oro[1], Sonia de Castro[4], Sonsoles Velázquez[4], Thomas P. Kohler[2], Elisabet Jiménez[5], Ana Medina[5], Isabel Usón[5,6], Lance E. Keller[7,8], Jessica L. Bradshaw[9¤], Larry S. McDaniel[7,8], María-José Camarasa[4], Uwe Völker[3], Sven Hammerschmidt[2]*, Juan A. Hermoso[1]*

1 Department of Crystallography and Structural Biology, Institute of Physical-Chemistry "Blas Cabrera", Spanish National Research Council (CSIC), Madrid; Spain, 2 Department of Molecular Genetics and Infection Biology, Interfaculty Institute for Genetics and Functional Genomics, Center for Functional Genomics of Microbes, University of Greifswald, Greifswald; Germany, 3 Department of Functional Genomics, Interfaculty Institute for Genetics and Functional Genomics, Center for Functional Genomics of Microbes, University Medicine Greifswald, Greifswald; Germany, 4 Instituto de Química Médica (IQM-CSIC), Madrid; Spain, 5 Crystallographic Methods, Institute of Molecular Biology of Barcelona (IBMB-CSIC), Barcelona Science Park, Helix Building, Barcelona; Spain, 6 ICREA, Institució Catalana de Recerca i Estudis Avançats, Barcelona; Spain, 7 Center for Immunology and Microbial Research, University of Mississippi Medical Center, Jackson, Mississippi; United States of America, 8 Department of Cell and Molecular Biology, University of Mississippi Medical Center, Jackson, Mississippi; United States of America, 9 Department of Microbiology and Immunology, University of Mississippi Medical Center, Jackson, Mississippi; United States of America

¤ Current address: Department of Pharmaceutical Sciences, University of North Texas Health Science Center, Fort Worth, Texas; United States of America
* sven.hammerschmidt@uni-greifswald.de (SH); xjuan@iqf.csic.es (JAH)

**Data Availability Statement:** Atomic coordinates for the reported crystallographic structures have been deposited at the Protein Data Bank (PDB;

## Abstract

ATP-binding cassette (ABC) transport systems are crucial for bacteria to ensure sufficient uptake of nutrients that are not produced de novo or improve the energy balance. The cell surface of the pathobiont *Streptococcus pneumoniae* (pneumococcus) is decorated with a substantial array of ABC transporters, critically influencing nasopharyngeal colonization and invasive infections. Given the auxotrophic nature of pneumococci for certain amino acids, the Ami ABC transporter system, orchestrating oligopeptide uptake, becomes indispensable in host compartments lacking amino acids. The system comprises five exposed Oligopeptide Binding Proteins (OBPs) and four proteins building the ABC transporter channel. Here, we present a structural analysis of all the OBPs in this system. Multiple crystallographic structures, capturing both open and closed conformations along with complexes involving chemically synthesized peptides, have been solved at high resolution providing insights into the molecular basis of their diverse peptide specificities. Mass spectrometry analysis of oligopeptides demonstrates the unexpected remarkable promiscuity of some of these proteins when expressed in *Escherichia coli*, displaying affinity for a wide range of peptides. Finally, a model is proposed for the complete Ami transport system in complex with its various OBPs. We further disclosed, through in silico modelling, some essential structural changes facilitating oligopeptide transport into the cellular cytoplasm. Thus, the structural analysis of

https://www.rcsb.org) under accession numbers 8A42, 8QLC, 8QLG, 8QLH, 8QLJ, 8QLK,8QLM, 8QLV and 8QMO. The Mass-Spectrometry raw data have been uploaded to ProteomeXChange (https://massive.ucsd.edu/ProteoSAFe/static/massive.jsp)) and are fully available (identifier: MSV000092727). All data presented are available in the main text or the Supplementary Materials.

**Funding:** This work is supported by grants PID2020-115331GB-100 funded by MCIN/AEI/10.13039/501100011033 and CRSII5_198737/1 by the Swiss National Science Foundation to J.A.H. Institutional funds University of Mississippi Medical Center to L.S.M. This work was further supported by the Bundesministerium für Bildung und Forschung (BMBF/DLR) - project Pneumofluidics – FKZ 01DP19007 (to S.H.) and BMBF-Zwanzig20 - InfectControl 2020 - projects VacoME - FKZ 03ZZ0816A and Pneumowiki – FKZ 03ZZ0839B (to S.H.). The funders had no role in study design, data collection and analysis, decision to publish, or preparation of the manuscript.

**Competing interests:** The authors have declared that no competing interests exist.

the Ami system provides valuable insights into the mechanism and specificity of oligopeptide binding by the different OBPs, shedding light on the intricacies of the uptake mechanism and the in vivo implications for this human pathogen.

## Author summary

The uptake of diverse oligopeptides enables pneumococcal growth despite auxotrophies and functions as a critical sensor for assessing the composition of the local environment. The identification of additional OBPs in non-encapsulated *S. pneumoniae* strains suggests their involvement in sensing a broader spectrum of bacterial competitors coexisting with the highly commensal pneumococcus. This study presents a comprehensive analysis of the initial phase of peptide transport mediated by OBPs within the pneumococcal Ami permease system. We disclose a common mechanism for oligopeptide recognition that is modulated in each OBP to accommodate a diverse array of oligopeptides. Understanding how pneumococcus perceives external stimuli and responds to them is imperative for unraveling the transition from a commensal to a pathogenic state.

## Introduction

The regulated uptake of molecules by cells is a fundamental biological process essential for all living organisms. This process enables cells to acquire nutrients from the surrounding medium and facilitates communication with their environment. Bacteria employ various importers and permeases to take up diverse molecules, a subset of which belongs to the extensive family of transport systems referred to as the ATP-binding cassette (ABC) transporters. *Streptococcus pneumoniae* (the pneumococcus) is a Gram-positive bacterium assuming both commensal and opportunistic pathogenic roles. Notably, pneumococci colonize the human nasopharyngeal cavity before causing invasive infections [1]. Successful colonization of the nasopharyngeal cavity necessitates the acquisition of amino acids from this dynamic niche. Specifically, pneumococcus is auxotrophic for cysteine, glycine, valine, leucine, isoleucine, arginine, asparagine, histidine, and glutamine [2–6]. Consequently, oligopeptide uptake from the milieu represents a pivotal mechanism for sustaining pneumococcal nutrition. Moreover, Gram-positive bacteria frequently employ oligopeptides as sensory molecules, capable of modulating the expression of numerous virulence factors, including hemolysins, adhesins, and genes mediating biofilm formation [7]. In particular, the pneumococcus recognizes and cooperatively responds to neighboring bacteria within its intricate ecological niche by sensing released peptide fragments [8–10].

The uptake of oligopeptides by pneumococci is performed by a specific ABC transporter system known as '*Ami*' [11]. All ABC transporters share a common architecture, comprising two permease components and two nucleotide-binding components (ATPases) [12]. Additionally, ABC import systems may involve a globular protein termed substrate-binding protein (SBP), serving as the initial receptor for the cargo. In Gram-positive species, the SBP is retained either anchored to the cell membrane *via* a lipid modification on the N-terminal cysteine or covalently linked to the translocation pore [13]. Crucially, SBPs determine the specificity of their respective transport systems, thereby governing the range of molecules capable of entering the cell. Despite exhibiting limited sequence similarity, these proteins share common features. They are bilobate molecules typically consisting of two domains, interconnected by a hinge region. The ligand binds within a deep cleft formed between both lobes [reviewed in [14] and are trapped by the so-called "*Venus fly-trap*" mechanism [15].

The Ami system was the first ABC transporter identified in Gram-positive bacteria [16]. It serves as the exclusive system responsible for the uptake of oligopeptides from the environment to overcome the pneumoccoccal auxotrophies [11]. The constituents of the Ami system include two transmembrane proteins, AmiC (SP_1890/ SP_RS09390 in *S. pneumoniae* TIGR4) and AmiD (SP_1889/ SP_RS09385), which collectively form a channel facilitating the selective passage of substrates across the membrane. In addition, two ATPases, namely AmiE (SP_1881/ SP_RS09380) and AmiF (SP_1887/ SP_RS09375), are positioned at the cytosolic face of the membrane, orchestrating substrate transport through the ATP hydrolysis. The Ami system is completed by the presence of the SBP AmiA (SP_1891/ SP_RS09395), which is an oligopeptide-binding lipoprotein (OBP) [17]. Following analyses unveiled that the Ami permease comprises four additional OBPs: AliA (SP_0366/ SP_RS01790), AliB (SP_1527/ SP_RS07525), AliC and AliD, all of which are paralogs of AmiA ('Ali' derives from '*Ami-like*') [18]. It is worth to mention that AliC and AliD are exclusively identified in non-encapsulated *S. pneumoniae* (NESp) [18] and are related with nasopharyngeal colonization and disease [19].

According to the Pfam database [20], the OBPs AmiA, AliA, AliB, AliC and AliD belong to the family SBP_bac_5 (PF00496). The N-terminus of these lipoproteins encompasses a signal peptide which includes the lipobox motif [L][AST][A][C] (**Fig 1A** and **1B**), that facilitates the translocation across the cytoplasmic membrane [21] and subsequent anchoring to the membrane by lipidation of the conserved cysteine residue (Cys23).

OBPs in the Ami system present molecular masses ranging from 53 to 70 kDa and belong to the cluster C of SBP [22], which includes SBPs with an additional third domain located within one of the two lobes.

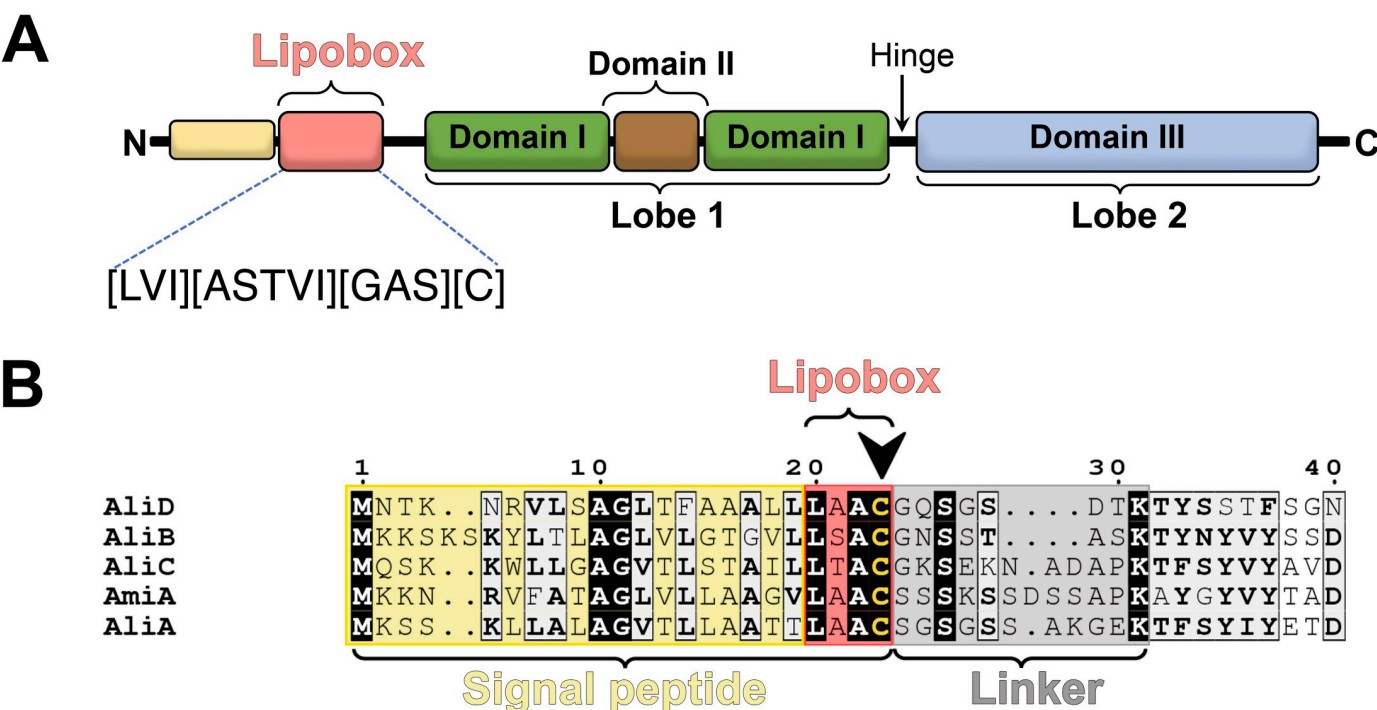

**Fig 1. Domain organization of the OBPs found in the Ami system in *S. pneumoniae*.** (**A**) Schematic representation illustrating the different elements composing the lipoproteins of the Ami system. Distinctive coloring is applied to each domain. (**B**) Alignment of sequences encompassing the signal peptide (in yellow), the lipobox (in red) and the linker region (in gray) in the five lipoproteins of the Ami system. The cysteine residue designated for lipidation is denoted by a black arrow.

The Ami system has been proposed to play a significant role in environmental signaling, the initiation of competence [11], as well as contributing to the successful colonization of the nasopharynx [23]. AliB, but not AmiA or AliA, is associated with meningitis [24]; suggesting that AliB may possess a distinct oligopeptide substrate specificity that cannot be compensated by AmiA or AliA. Notably, in the absence of capsular polysaccharides and PspK, group II NESp strains harboring *aliC* and *aliD* within the deleted *cps* locus, are still capable of nasopharyngeal colonization [9,25–27]. Recent investigations led to the conclusion that AliC enhances pneumococcal competence while AliD facilitates early nasopharyngeal colonization of the mouse [9] both are crucial in persistence during invasive disease [28]. Moreover, AliC and AliD bind peptides matching ribosomal proteins of other bacterial species, indicating a potential route for interspecies communication within the nasopharynx [9]. Subsequent investigations revealed a similar behavior for AmiA, AliA and AliB [10].

In this work, we present the atomic-resolution structural characterization of the OBPs from the Ami system that illuminate the dynamic structural changes occurring between open and closed conformations. Mass spectrometry experiments using samples derived from AliB and AmiA, revealed a considerable abundance of peptides sequestered during heterologous expression. Analysis of these peptides, together with structures of the OBP-peptide complexes reported here, provides a detailed comprehension of the specific peptide recognition patterns intrinsic to the OBPs of the pneumococcal Ami system.

## Results

### Structural determination of OBPs within the Ami system. Three-dimensional insights into the apo conformations of AliD and AliC

The pneumococcal OBPs belonging to the Ami system, namely AmiA (24–659), AliA (residues 24–660), AliB (residues 26–652), AliC (residues 24–655) and AliD (residues 28–652), were expressed and purified deprived of the lipobox segment (see Methods). Next, all these OBPs, with the exception of AliA, were crystallized leading to the resolution of their three-dimensional structures (S10 Table). In addition, synthesis of five oligopeptides (peptides **1–5**) was undertaken for this work (**Table 1**).

Specifically, nine distinct crystal structures were resolved, representing the apo state (open conformation) or complexed (holo state) with the indicated synthesized peptides, showcasing closed conformations. The resolutions of these structures ranged from 1.49Å to 2.38Å (**Tables 1** and **S10**).

The five lipoproteins exhibit pairwise sequence identity of ~52–60% (**S1 Table** and **S1 Fig**) and share a very similar overall folding. Consequently, the 1.8 Å-resolution open conformation

**Table 1. Crystal structures of Oligopeptide-Binding Proteins (OBP) of the Ami system.**

| OBP | Conformation | Oligopeptide | Oligopeptide Sequence | Resolution (Å) |
|------|--------------|--------------|------------------------|----------------|
| **AmiA** | Closed (AmiA-1) | Yes | Unknown | 1.50 |
| | Closed (AmiA-2) | Yes | Peptide **5** (AKTIKITQTR) | 1.76 |
| **AliB** | Closed (AliB-1) | Yes | Unknown | 1.65 |
| | Closed (AliB-2) | Yes | Peptide **2** (AIQSEKARKHN) | 2.29 |
| | Closed (AliB-4) | Yes | Peptide **3** (PIVGGHEGAGV) | 1.66 |
| | Closed (AliB-3) | Yes | Peptide **4** (VMVKGPGPGREST) | 1.49 |
| **AliC** | Open (AliC-1) | No | - | 2.38 |
| **AliD** | Open (AliD-1) | No | - | 1.80 |
| | Closed (AliD-2) | Yes | Peptide **1** (FPPQSV) | 2.18 |

of AliD was selected as a representative structure to describe the global architecture of OBPs in the Ami system. Moreover, AliD was the only OBP for which we could obtain both conformations (open and closed, see below). Unless otherwise specified, structural features observed in AliD can be extrapolated to the other family members. The obtained electron density map of the AliD structure, encompassing residues 28 to 652, presents an excellent quality all over the sequence (S2A Fig). AliD adopts a bilobate structure characterized by three domains interconnected by short polypeptide chain segments serving as hinges (Fig 2).

The two lobes enclose a deep cavity measuring ~16 Å in width and 45 Å in height, housing the peptide-binding pocket (Fig 2B). Lobe 1 comprises Domain I (residues 29–63, 191–296, and 583–617) and Domain II (residues 64–190), while Lobe 2 consists solely Domain III (residues 297–582) (Fig 2A). This structural organization, previously observed in other OBPs such

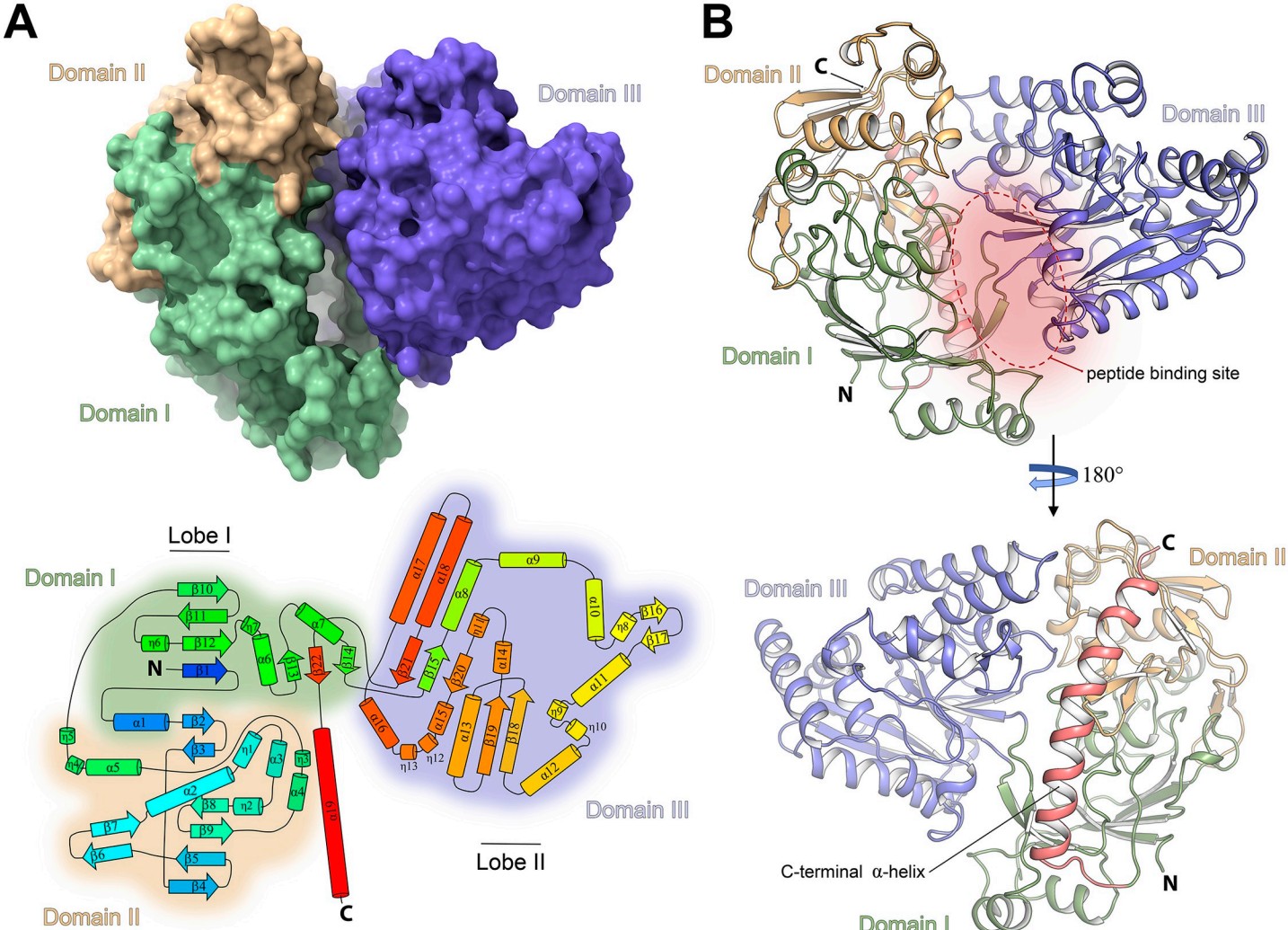

**Fig 2. Crystal structure of AliD in the open conformation.** (**A**) Molecular surface and topology of AliD. The upper panel illustrates the molecular surface representation of AliD, with each domain color-coded. In the lower panel, a topological diagram depicts the secondary structure elements of AliD, labeled and numbered. The color code spans from blue (N-terminus) to red (C-terminus), portraying α-helices as cylinders and β-strands as arrows. (**B**) Overall cartoon representation of AliD monomer structure. The structure is presented in two orientations spaced 180˚ apart. Domains I, II, and III are labeled and colored green, light orange, and blue, respectively. The dimensions of the peptide-binding cavity (~16 Å in width and ~45 Å in height) are specified. The C-terminal α-helix (α19) is highlighted in red and labeled. N, amino-terminus; C, Carboxy-terminus.

as OppA from *Salmonella typhimurium* and *Lactococcus lactis* [29,30], is implicated in the transport of peptides ranging from two to five residues in length, including cell-wall peptides containing γ-linked and D-amino acids [31]. Although AliD and OppA [PDB 1RKM [32], with a sequence identity of 25.76%, **S3B Fig**] share a similar overall structural organization, notable differences exist between them, as evidenced by the high *rmsd* values obtained upon superposition (~3.2 Å, **S3A Fig**). Nevertheless, both proteins display a mixed β-sheet arrangement within each lobe, where strands undergo a characteristic cross-over pattern involving the so-called regions c1 and c2 (**S3A Fig**).

The most striking feature of the AliD structure compared to OppA is the presence of an intriguing additional C-terminal 30-aa long α-helix (α19, spanning residues 622–652; **Fig 2A**). This α-helix, which serves as distinctive hallmark of all the Ali proteins, establishes numerous hydrophobic and polar interactions with Domains I and II from Lobe 1, contributing significantly to the structural stability of the protein. The exposed surface of α19 is decorated with multiple basic residues (**S4 Fig**), a conserved feature across all the OBPs in the Ami system (**S1 Fig**) that, besides its clear structural role, could help in proper orientation of the OBP on the membrane during oligopeptide binding.

AliC was also crystallized in the open conformation, and its structure manifests all the characteristics delineated for AliD. Remarkably, the crystal structure of AliC unveils a distinctive three-dimensional domain-swapped dimer arrangement. In this configuration, the Lobe 1 (comprising Domain I and II) of one monomer becomes intertwined with the Lobe 2 (Domain III) of the adjacent monomer, resulting in an overall architecture that is identical for the two functional monomeric units (**S5A Fig**). The resulting entwined dimer adopts an open conformation, but the width of the ligand-binding cleft is expanded compared to that observed in AliD, measuring 29 Å instead of 16 Å. Consequently, AliC exhibits a 13 Å wider open conformation (**S5B Fig**). The observed domain swapping in AliC would be compatible with its insertion into the membrane, given that both N-terminal ends would be oriented toward the same side. This phenomenon suggests a potential additional regulatory mechanism detected in AliC but not occurring in the other analyzed OBPs. This dimeric nature of AliC (experimentally observed also in solution by analytical ultracentrifugation technique, **S5C Fig**) is intriguing because these OBPs, especially when associated with the ABC transporter, are believed to be monomeric. A swapped dimer structure has also been observed in the α-keto acid SBP protein TakP from *Rhodobacter sphaeroides* [33] and in the carbohydrate SBP SP_0092 (SP_RS00475) from *S. pneumoniae* [34]. Nevertheless, to date, there is no supporting evidence indicating that a domain-swapped dimer represents a functional state of these SBPs, including AliC.

### Three-dimensional structure of the AliD:peptide 1 complex

To investigate the conformational changes occurring in the Ami permease OBPs upon ligand binding, we synthesized specific peptides (**Tables 1** and **S2**) previously reported to be substrates of these proteins [9,10]. One such peptide, FPPQSV (peptide **1**), derived from *Prevotella* species, has been shown to induce significant changes in gene expression and enhanced colonization when bound by AliD [9]. To verify specific binding of peptides to OBPs, binding of peptide **1** (FPPQSV), which is the natural substrate of AliD, and peptide **2** (AIQSEKARKHN) was measured to AliD by microscale thermophoresis (MST) (see methods). Both peptides showed a concentration dependent binding to AliD (**S6A Fig**) with calculated dissociation constants in the μM-range. The Kd for peptide **2** is 14.4 μM and peptide **1**, the natural substrate of AliD, has a Kd of 3,44 μM. Hence, peptides bind to OBPs, however, peptide-binding occurs with a low affinity when both, the fluorescently labelled AliD and the peptides as ligand are in solution and binding experiments are conducted at 25˚ C. The crystal structure of the AliD:peptide

**1** complex was obtained by co-crystallization (see Materials and Methods) at 2.18 Å resolution (**Fig 3A**).

The asymmetric unit contained two monomers (chains A and B) with virtually identical structures (*rmsd* of 0.14 Å for 564 Cα atoms superposition). In these structures, AliD mono- mers adopt a closed conformation, resulting in a complete engulfment of peptide **1** between the two protein lobes without contact with the bulk solvent (**S1 Movie** and **Fig 3A**). The ligand-bound and unbound forms of AliD retained the three-dimensional structures of the individual domains unaltered, as both conformations were related by a rigid-body rotation of ~20˚ of Lobe 1 (Domain III) relative to Lobe 2 (Domains I and III) (**S1 Movie**). The peptide- binding site is located in the crevice between the two lobes, akin to other substrate-binding proteins [32]. The electron density for the AliD:peptide **1** structure was of high quality, enabling straightforward model building of the hexapeptide FPPQSV in both chains of the asymmetric unit (**S6 and S7 Figs**) and facilitating the identification of the interactions that sta- bilize the substrate within AliD (**Fig 3A**). In the complex, binding of peptide **1** is facilitated through specialized pockets within the structural framework of AliD that accommodate the diverse side chains of the peptide. These protein pockets, denoted as P1, P2. . ., and so forth; each play a distinct role in accommodating the lateral chain of corresponding oligopeptide res- idues (namely residue 1, 2,. . .). In-depth description of the constituent residues comprising each of the AliD pockets involved in the recognition of peptide **1** detailed in **S3 Table**. This analytical approach has been systematically applied to the other OBP-peptide complexes here reported (see below). Specifically, the side chain of the initial substrate residue (F) finds accom- modation within a hydrophobic pocket cavity, designated as P1. This pocket is delineated by the spatial arrangement of residues Y50, A52, K297, V499 and I603 (**Fig 3A**). The amine N-ter- minal tip of the oligopeptide substrate establishes electrostatic interactions with residue D501 and forms H-bonds with the main-chain atoms of residue V499, both from Domain III (**Fig 3A**). Residue D501 assumes a crucial role in the interaction by also establishing H-bond inter- actions with residue W498, which is involved in clamping the first two substrate residues through Van der Waals interactions (**Fig 3A**). Moreover, residues located in Domain I of AliD including G39, A52, D53 and N608 (located in pockets P4 and P5, **S3 Table**), actively contrib- ute to stabilizing the substrate through polar interactions (**Fig 3A**). These polar interactions are complemented by several hydrophobic interactions facilitated by aromatic residues such as Y50, Y51, W498, F481, F518 and Y519 (located in pockets P1-P3, **Fig 3A** and **S3 Table**), result- ing in the tight binding of the initial five residues of peptide **1** to the AliD protein. In contrast, the ultimate residue of peptide **1** demonstrate fewer interactions with the OBP. Consistently, we observed an increase in the B-factors of the substrate from the N- to C-terminus, along with the presence of two different conformations for the sixth residue in the substrate (**S6 Fig**).

## Substrate recognition by other OBPs of the Ami system

**Substrate binding in AliB.** The crystal structure of AliB (residues 28–652) was solved at 1.65 Å resolution, revealing a closed conformation despite the absence of pre-incubation with any peptide. During the modeling and refinement process, additional electron density at the AliB substrate-binding site became evident (**S8 Fig**), likely originating from *Escherichia coli* peptides bound during the purification process. Thus, we modeled *E. coli* peptides (VMVKGPGPGREST and PIVGGHEGAGV) previously reported to bind AliB [10]. Although the electron density map clearly outlined the backbone of a nonapeptide, the side-chain den- sity exhibited ambiguity in places, suggesting the binding of a heterogeneous population of peptides (**S8 Fig**). Given that the observed peptide lengths did not align with the reported *E. coli* peptides, an identification effort was undertaken through mass spectrometry (see below).

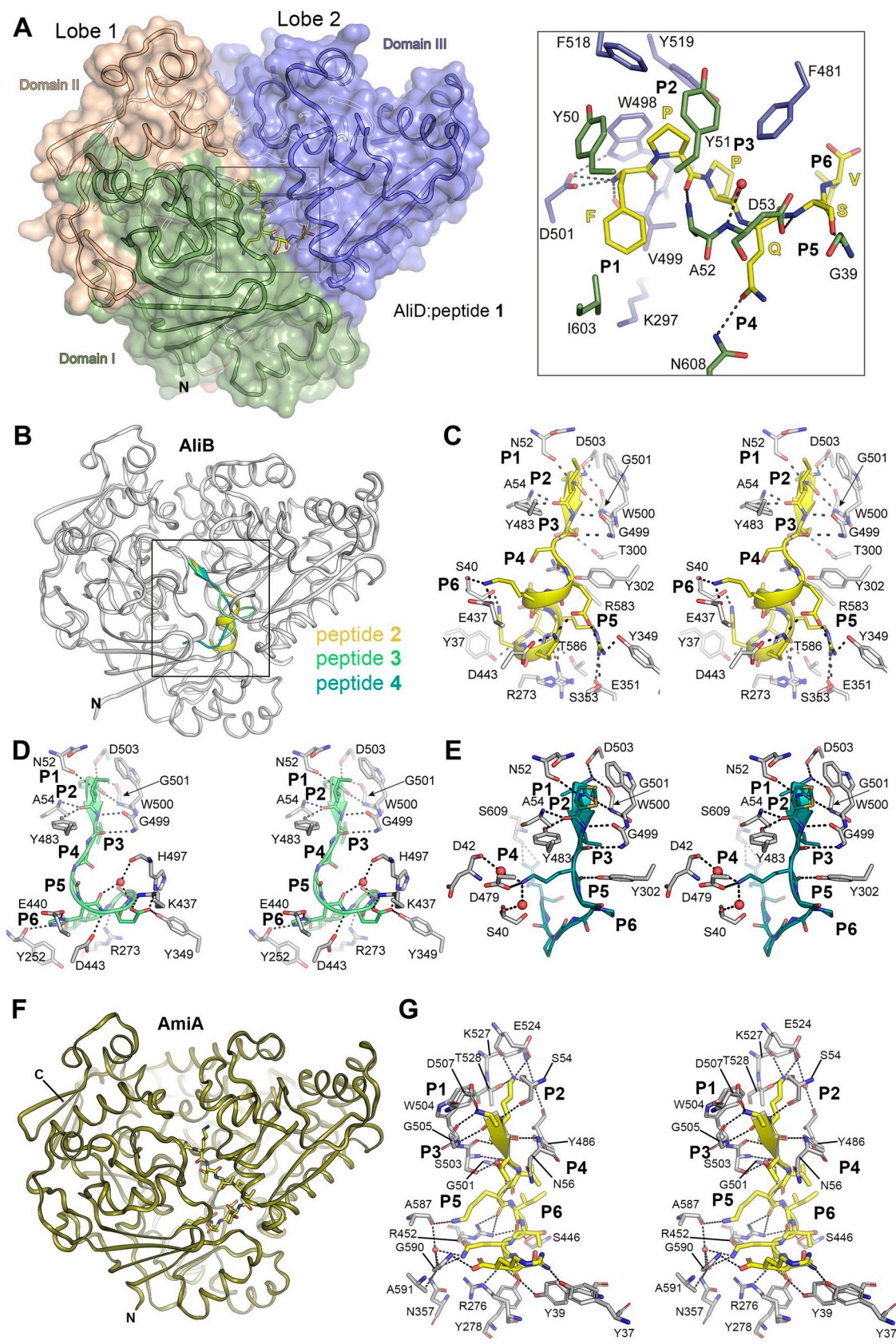

**Fig 3. Substrate recognition in AliD, AliB and AmiA.** (**A**) Crystal structure of the AliD:peptide **1** complex. Left, ribbon representation of the AliD:peptide **1** complex (ligand shown in yellow caped sticks). Domains I, II, and III are labeled and colored green, light orange and blue; respectively. Right, a magnified view (corresponding to the boxed area shown in the left panel) highlighting amino acid residues engaged in the recognition of peptide **1** by AliD. Polar interactions are indicated by dotted black lines. The sequence of peptide **1** (FPPQNV) is illustrated in yellow capital letters. (**B**) The crystal structure of AliB in complex with peptides **2**, **3** and **4**. AliB, as observed in the AliB:peptide **2** complex, is depicted as a white cartoon. The substrates are represented with their secondary structure elements and are color-coded as yellow (peptide **2**), green (peptide **3**) and blue (peptide **4**). (**C**) Stereo view illustrating polar interactions between AliB (white sticks) and peptide **2** (yellow sticks). (**D**) Stereo view showcasing polar interactions between AliB (white sticks) and peptide **3** (green sticks). (**E**) Stereo view illustrating polar interactions between AliB (white sticks) and peptide **4** (blue sticks). (**F**) Ribbon representation of AmiA in complex with peptide **4** (depicted as yellow capped sticks). (**G**) Stereo view providing specific details of peptide **4** recognition by AmiA. The relevant residues and crystallographic water molecules are depicted as gray-capped sticks and red spheres, respectively. Residues participating in hydrophobic interactions have been omitted for clarity. P1 to P6, pocket 1 to pocket 6. Pockets beyond substrate position 6 have been omitted for clarity.

To evaluate AliB's capacity to recognize diverse peptides, the protein was incubated with an excess of defined synthesized peptides (peptides **2**, **3** and **4**) that were previously reported to bind AliB [10]. This approach facilitated the determination of crystal structures of AliB in complex with these three different peptides (**S2** and **S10 Tables**). Specifically, crystal structures of AliB in complex with peptide **2** (AIQSEKARKHN), peptide **3** (PIVGGHEGAGV), and peptide **4** (VMVKGPGPGREST) were solved at resolutions of 2.29 Å, 1.66 Å and 1.49 Å, respectively (**S10 Table**). These structures offer a unique description of AliB's capability to recognize oligopeptides of varying sizes and composition (**Fig 3B**). **S4**, **S5**, and **S6 Tables** provide a comprehensive breakdown of the AliB residues constituting the various pockets implicated in the recognition of the side chains of peptides **2**, **3**, and **4**, respectively. While the overall closed conformations of AliD and AliB exhibit significant similarity (*rmsd* 0.88Å for 588 Cα atoms superposition; **S9 Fig**), two key aspects differ: (*i*) the arrangement of the Lobes, wherein AliB adopts a more tightly closed conformation compared to AliD, and (*ii*) the conformation of specific loops shaping the substrate-binding site at positions 1–4. As detailed later, these distinctions, coupled with the specific amino acid residues shaping the binding site, likely contribute to the observed variations in substrate specificity between these two OBPs.

The three complexes obtained for AliB exhibited minimal backbone changes among them (*rmsd* ranging between 0.1 and 0.3 Å). In all three complexes, mirroring the observation in the AliD:peptide **1** complex, the amine N-terminus of the oligopeptide establishes a salt-bridge with an aspartic residue (D503) and van der Waals interactions with the neighboring tryptophan residue (W500) (**Fig 3C**–**3E**). The region encompassing the first three residues of the ligands (AIQ in peptide **2**, PIV in peptide **3** and VMV in peptide **4**) folds in an antiparallel β-sheet-fashion with the β20 (G499, W500 and G501) from Lobe 1, and parallel β-sheet-like interactions with a loop of domain I (containing residues N52 and A54) on the opposite side (**Fig 3C**–**3E**). This recognition pattern for the initial three residues of the peptide is conserved in both AliD and AliB complexes and, as detailed below, is also observed in the AmiA complex. Given the conservation of the structure and key residues in this region (involving pockets P1, P2 and P3), the recognition pattern for the first three residues of the substrate could be considered a hallmark of all OBPs within the Ami permease system. Beyond the third position, the recognition by AliB does not adhere to a conserved pattern, as each peptide adopts a distinct conformation within the substrate-binding cavity and establishes different interactions with the protein (**Fig 3C**–**3E**). For instance, in the case of peptide **2,** the remaining portion of the oligopeptide immediately following the small β strand region, which exhibits a strong basic character (EKARKHN), adopts an α-helix conformation within the AliB cavity (**Fig 3C**). This helical region finds stabilization through numerous H-bonds with the protein and is further reinforced by salt-bridge interactions involving the basic residues of the substrate interacting with the acidic residues of the protein (E351, E437, and D443) (**Fig 3C** and **S4 Table**).

Peptide **3**, comprising 11 residues (PIVGGHEGAGV), shares the same length as peptide **2**, but possesses a distinctly different sequence. Analogous to peptide **2**, the initial three residues adopt a short β-strand conformation. However, in contrast to peptide **2**, the subsequent portion of peptide **3** lacks a substantial quantity of basic residues, comprising several glycines and other non-bulky residues. In this case, the ligand inside the AliB binding-site does not exhibit any secondary structural elements except for the conserved β-strand at the N-terminus (**Fig 3D**). The two glycine (G) residues in the peptide (residues 4 and 5) do not engage in polar interactions with AliB, while the following two G residues (residues 8 and 10) establish strong stabilizing H-bonds. Particularly noteworthy are the salt-bridge interactions that take place between peptide **3** and AliB (H6 from the substrate with E440 from AliB, and E7 from the substrate with K437 and H497 from AliB) (**Fig 3D** and **S5 Table**). Additionally, a few other H-bond contacts are observed with the last four residues of the substrate (GAGV).

The last AliB complex, AliB:peptide **4** (VMVKGPGPGREST), exhibits a recognition pattern akin to the description provided above for the initial three residues of the ligand. Nevertheless, two notable distinctions emerge compared to the previous cases. Firstly, a *cis* peptide bond is present at position 6 of the peptide, corresponding to a proline. Secondly, the electron density representing the last three peptide residues (EST) weakens significantly, indicating a lack of stabilization for the C-terminus of the ligand (**Fig 3E**). Detailed composition for each substrate-binding pocket in AliB:peptide **4** complex is provided in **S6 Table**.

**Substrate binding in AmiA.** As we observed in the case of AliB, the analysis of AmiA structure (solved at 1.50 Å resolution) revealed an unidentified electron density trapped at the substrate-binding site (**S10A Fig**). Considering the precedent of this observation in previous reports for AmiA and the identification of the peptide's nature [10], we modeled this decapeptide reasonably matching the observed experimental map (**S10B Fig**). To mitigate uncertainties stemming from potential mixtures of different peptides in our crystallized structure, we conducted pre-incubation of the AmiA protein with an excess of synthetized peptide **5** before crystallization. This strategy facilitated the elucidation of the AmiA:peptide **5** complex at 1.76 Å resolution (**Fig 3F**). Four independent monomers were found in the asymmetric unit, all with a highly similar structure (*rmsd* value ranging from 0.14 to 0.21 Å for 574 Cα atoms among the four monomers, using monomer B as a reference throughout the text). Furthermore, the peptide **5** in the four chains exhibits an identical conformation of the peptide backbone (**S11 Fig**). In this complex, the amine N-terminus of peptide **5** is stabilized in a manner analogous to the description provided above for the other complexes, involving pockets P1, P2 and P3 from the OBP (**Fig 3G** and **S7 Table**). It forms a salt-bridge with an aspartic residue (D507) and engages in hydrophobic interactions with a neighboring tryptophan residue (W504). The initial four residues of the peptide (AKTI) establish antiparallel β-sheet-like interactions, engaging the terminal residues of the extended strand of β20 on one side and forming parallel β-sheet-like interactions with a loop of domain I (which includes residues S54 and N56, from pockets P2 and P3, respectively), on the opposite side. The oligopeptide adopts an extended conformation with minimal intramolecular interactions, displaying a kink of ~90° at residue 6, resulting in an "L" shape. The two isoleucine residues in peptide **5** (I4 and I6) tightly pack against each other, occupying the two hydrophobic pockets P4 (involving residues Y486 and V502) and P6 (formed by residues D443 [aliphatic chain], S446, Y449, F481 and L482), all situated in AmiA's domain III. The lysine residue at position 2 (K2) of the oligopeptide is stabilized through a salt-bridge interaction with E524 and a H-bond with T528 (domain III, both residues located in P2) of AmiA. The remaining charged side chains of the oligopeptide (K5, Q8, and R10) establish multiple hydrogen bonds with specific AmiA residues; notably, in the case of Q8, mediated by water molecules (**Fig 3G**). The C-terminus of the peptide finds

stabilization through hydrogen bonding with the nitrogen backbone atom of G590, located at the interphase between domains I and III.

**Mass Spectrometry Analysis of the oligopeptides captured by AliB and AmiA during purification.** The three-dimensional structures of AmiA and AliB revealed the capture of oligopeptides during their expression and purification from *E. coli*. To determine the identity of these bound substrates, we isolated the peptides bound to AmiA and AliB following established protocols [9] and subjected them to liquid chromatography followed by mass spectrometry (LC-MS) analysis (see Methods). For identification criteria, we considered only peptides whose spectra with +2H charge state had been detected at least twice. Surprisingly, our analysis identified a total of 356 distinct peptides bound to the AmiA protein and 393 peptides bound to AliB. Notably, both AmiA and AliB bound large peptides ranging from 8–16 amino acids (**S12A Fig**). AmiA predominantly recognized peptides consisting of 10–11 residues, while AliB favored 11–12 residues (**S12A Fig**). AmiA and AliB showed a tendency to bind peptides with a hydrophobic amino acid, particularly valine, as the first residue. As expected by the large number of oligopeptides trapped in both OBPs, sequence analysis (**S12B Fig**) reveals that there is not a consensus sequence for the 16 (AliB), or 11 (AmiA), residues bound. Even for the first five to six initial positions of the oligopeptides, the analysis shows tendencies but not a strict sequence consensus for oligopeptides bound to AmiA nor to AliB (**S12B Fig**). In both proteins, the P1 pocket encompasses a capacious cavity featuring exposed hydrophobic residues alongside others conferring a polar character. This configuration permits peptides accommodated by both, AliB and AmiA, to possess a variable lateral chain at position 1, exhibiting diverse sizes and physicochemical properties (*e.g.* E, I, M, Y, K). A parallel situation is observed in the P2 pocket of both proteins; however, in the case of AliB, there is a more pronounced inclination to accommodate a polar chain such as E or Q. Concerning the P3 pocket, again both proteins can accommodate side chains of diverse nature at the ligand's position 3. Notably, AmiA exhibits a stronger propensity to accommodate a negatively-charged residue, such as D or E, at this ligand position (**S12B Fig**), a trend also observed for AmiA's ligand positions 4 and 5. Beyond these positions, there is a strong variability in the sequence of oligopeptides bound to AmiA, and to AliB. Remarkably, a considerable number of the identified peptides were found to originate from very abundant cytosolic (ribosomal and metabolic) proteins, aligning with prior observations [10]. Also, a significant proportion of the peptides belonged to proteins involved in processes not previously linked to this context, including transcription, DNA replication or macromolecule transport (**S12C Fig**). The distribution of peptides derived from proteins engaged in various processes exhibited some differences between AmiA and AliB. Specifically, AmiA exhibited a higher proportion of peptides derived from proteins involved in cellular metabolism compared to ribosomal proteins, while AliB demonstrated a more balanced distribution. Furthermore, AmiA recognized peptides derived from proteins involved in cellular processes that were not detected by AliB, and *vice versa* (**S12C Fig**). It is worth to mention that AliD and AliC, both of which were also expressed in *E. coli*, we did not observe peptides originating from *E. coli* within the crystal structures of these proteins. This observation highlights distinctive peptide recognition patterns among the OBPs of the pneumococcal Ami system.

## Discussion

### Molecular basis of substrate recognition by OBPs in the Ami System

The structural elucidation presented herein encompasses four of the five OBPs within the Ami system, (in both apo and holo states with diverse oligopeptides) and provides insights into substrate recognition and specificity mediated by these OBPs. Mass spectrometry analysis of

peptides trapped by AliB and AmiA during purification unveil an unexpected capacity of both OBPs to capture hundreds of distinct oligopeptides, with a maximum peptide length of 16 amino acids. This observation suggests a lack of stringent specificity in the nature of the trapped substrates (**S12B Fig**). Such promiscuity aligns with two key factors: (*i*) the substantial volume of the substrate-binding cavity in the Ami system OBPs (**S13A Fig**), which is the largest among OBPs within the same family (**S13B Fig**); and (*ii*) the primary involvement of the N-terminal region of the oligopeptide in substrate recognition (see below).

AliB was crystallized with three different peptides (**2**, **3**, and **4**), revealing that recognition primarily hinges on protein-oligopeptide interactions involving Lobe 2 of AliB with the N-terminal region of the oligopeptide (*i.e.*, salt-bridge interaction with D503, van der Waals interaction with W500, and main chain H-bond interactions of β20 with the initial three positions of the oligopeptide in an antiparallel β-strand fashion, **Figs 3** and **4A**).

Furthermore, a comparative analysis of the three complexes indicates that no substantial alterations in the backbone or in side chains of AliB are requisite for the recognition of these three peptides (**S14A Fig**). Consequently, the first three positions of the oligopeptides precisely aligned with the AliB binding site, whereas beyond the fourth position, different conformations of the oligopeptide are observed (**Fig 4A**). Regardless, the electrostatic potential of the AliB binding site exhibits a neutral character to facilitate interaction with the residue at position 2 through Van der Waals interactions, and a strongly acidic character after position 4, enabling the stabilization of basic residues of the oligopeptides through salt-bridge interactions (**Fig 4A**). Interestingly, while maintaining the interaction pattern observed in AliB, AmiA features a β20 that is half the size of AliB's (6 residues in AliB *vs* 3 residues in AmiA) (**Fig 4B**). This shorter β-sheet is preceded by a coil that extends interactions with the oligopeptide backbone to residue 4 (as opposed to residue 3 in AliB). Indeed, the interaction of AmiA with the oligopeptide backbone propagates to residue 6 through interactions with side chains within the AmiA binding-site. Moreover, distinct structural disparities between the binding sites of AliB and AmiA are evident in two regions of Lobe 2, denoted as R1 and R2 (**Figs 4A**, **4B,** and **S1**). AmiA features an additional α helix in R1 (residues 523–530), a characteristic absent in AliB, AliC and AliD, which instead exhibit a four-residues-long $3_{10}$ helix. In addition, AmiA presents an acidic residue that mediates a salt-bridge interaction with the Lys residue at position 2 of peptide **5** (**Fig 4B**). Regions R1 and R2 (residues 532–538) together form the pockets P2, crucial for stabilizing the side chain from residue 2 of the oligopeptide, while pockets P4 and beyond enable the stabilization of various conformations of the oligopeptide beyond position 4 (**Fig 4A** and **4B**). This combination of structural elements introduces notable differences in the shape and the electrostatic potential of the oligopeptide-binding cavity in both OBPs. Considering these structural characteristics, the five OBPs within the Ami system can be categorized into two groups: (*i*) the *AliB-like* group (exhibiting a 6 residues-long β20 and lacking an additional α-helix in R1), comprising AliB, AliC and AliD; and (*ii*) the *AmiA-like* group (featuring a 3 residues-long β20 and possessing an additional α-helix in R1), comprising AmiA and likely also AliA [as predicted by AlphaFold (AF)] (**Fig 4C**). These disparities in the secondary structural elements, along with the nature of the residues decorating P1 to P4 pockets, provide distinctive particular features to the binding sites of each OBP (**S14B Fig**). These features are expected to influence the range of oligopeptides recognized by each OBP within the Ami system, in conjunction with the potential to accommodate different peptides in distinct manners (involving different pockets), extending beyond the ligand's fourth position.

For instance, AliD presents a significant number of aromatic residues decorating the binding-site cavity (**Fig 3A**), promoting the binding oligopeptides with a high hydrophobic residue content. Both AliB and AliC share strongly acidic pockets P3 and P4 but AliC features distinctive basic pockets P1 and P2 (**S14 Fig**). This suggests that AliC may bind similar oligopeptides

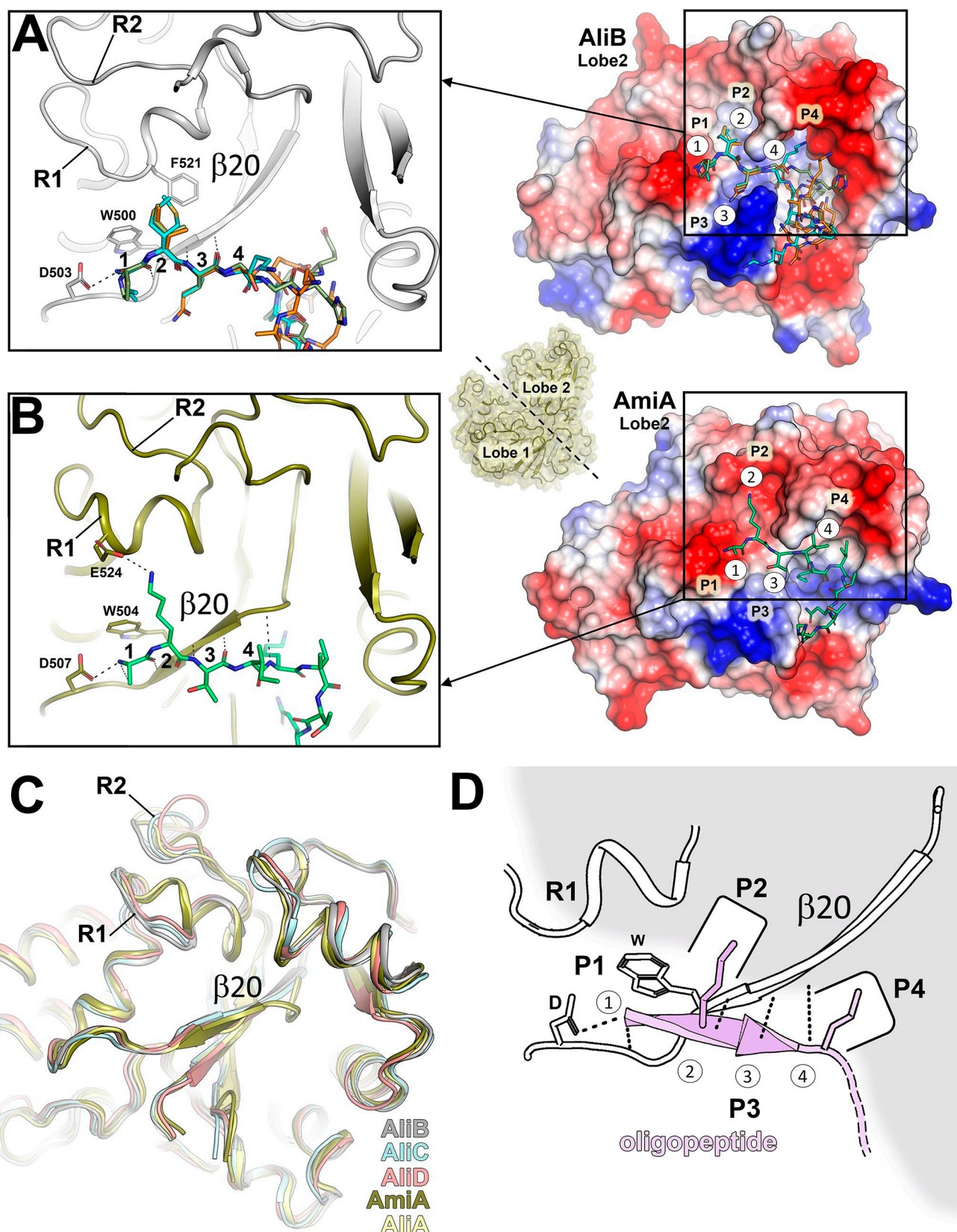

**Fig 4. Structural principles underlying peptide recognition by SBPs of the Ami permease in *S. pneumoniae*.** (**A**) Right panel, molecular electrostatic potential surface (MEPS) of AliB's lobe 2 is shown. The three peptides complexed with AliB are depicted as capped sticks with colors orange (peptide **2**), green (peptide **3**) and cyan (peptide **4**). The oligopeptide positions 1 to 4 and the pockets P1 to P4 are labeled. Left panel, cartoon representation corresponding to the boxed area displayed in the right panel. Conserved Aspartate (D) and Tryptophan (W) residues are depicted in capped gray sticks. The residue F521, crucial for stabilizing the lateral chain in oligopeptide position 2, is also shown as capped gray sticks. (**B**) Right panel, MEPS of AmiA's lobe 2 is shown. The peptide corresponding to the AmiA:peptide **5** complex is depicted in capped sticks (colored green). The oligopeptide positions 1 to 4 and the pockets P1 to P4 are labeled. Left panel, cartoon representation corresponding to the boxed area is in the right panel is displayed. Conserved Asp (D) and Trp (W) residues are depicted as capped deep olive sticks. The residue E524, critical for stabilizing the lateral chain in oligopeptide position 2, is also shown as capped gray sticks. Polar interactions are represented with dashed black lines. R1 and R2 represent variable regions 1 and 2, respectively. The small panel between panels A and B provides a view of a typical Ami OBP with both lobes indicated. The black dashed line between the two lobes indicates the position of the slice required to obtain the view of the MEPS of lobe 2 for both panels A and B. (**C**) Structural superposition of lobe 2 for the five Ami OBPs is presented, with a focus on strand β20. Regions R1 and R2 are indicated. (**D**) Schematic cartoon representing the relevant OBP structural features involved in oligopeptide recognition (colored in pink).

as AliB from position 4, but with a preference for acidic residues at position 2, as previously reported [9,10]. Within the *AmiA-like* group, AliA is expected to maintain a preference for a basic residue at position 2 (**S14B Fig**) while the observed basic potential in pockets P3 and P4, points the AliA capability to bind oligopeptides with acidic residues from position 4.

In summary, our structures reveal a common recognition pattern (**Fig 4D**) in which the backbone of the N-terminal residues of the substrate must form an anti-parallel β-strand interaction with β20 of Lobe 2 (encompassing the first three positions for the AliB group and four positions for the AmiA group). Substrate recognition also involves the side chain of the oligopeptide residue 2 within pocket P2. Additionally, the amine tip of the oligopeptide backbone forms a salt-bridge interaction *via* a conserved aspartate. The capacious cavities in the OBPs of the Ami system facilitate the stabilization of oligopeptides up to 16 residues in length. Contrary to the N-terminal region, these longer oligopeptides adopt different folds (as coils or even as α-helices, **Fig 3**) after position 4 and are stabilized by pockets beyond P4. Following recognition by Lobe 2, Lobe 1 further stabilizes the substrate through a few interactions with the N-terminal oligopeptide backbone and side-chain interactions with residues beyond the 9th position of the oligopeptide (**Fig 3**). This system, while adhering to a common mechanism of recognition, imparts considerable versatility to the five OBPs, enabling the recognition of different types of oligopeptides. This agrees with our MST measures showing a moderate binding affinity (Kd in the μM range) and not strong discrimination by the nature of the oligopeptide sequence [as reflected by the AliD ability to bind both peptide 1 and peptide 2 with similar affinities (**S6A Fig**)].

The large number of different peptides identified in AmiA and AliB is remarkable. Interestingly, most of these oligopeptides come from cytosolic proteins (ribosomal and metabolic proteins) that are only exposed after bacterial lysis. These *E. coli* proteins present high sequence identity with their homologs in *S. pneumoniae*. Thus, the lack of strong specificity for the substrate together with the cytosolic nature of the oligopeptides could be related to the well-known autolytic process in *S. pneumoniae*.

At the end of the competent state, pneumococci kill non-competent siblings by the release of specific toxins and activation of autolysins in non-competent pneumococci, a mechanism termed fratricide [35,36]. In this sense, the observed promiscuity of substrates in AmiA and AliB could provide, besides the DNA uptake associated with competence, a further advantage by the uptake of diverse oligopeptides containing residues for which *S. pneumoniae* is auxotroph (more than 90% of the oligopeptides detected in AmiA and AliB contained such residues).

## Oligopeptide transport by the Ami permease system

To investigate the potential interactions between these OBPs and their corresponding permease, as well as to elucidate the mechanism of oligopeptide transport, we employed

AF-Multimer [37] predictions for the entire permease system alone (AmiC, AmiD, AmiE and AmiF) and in complex with the five different OBPs (AmiA, AliA, AliB, AliC and AliD).

The resulting models displayed a high degree of similarity and exhibited very high confidence levels both for the complete permease system and the OBPs, particularly those with experimental structures reported here (**S15 Fig**). In all cases, the interaction between the OBP and the permease occurred in a remarkably similar manner, all the OBPs are in its closed conformation and involving the same specific regions of both the permease and the OBP. Consequently, we will focus on the particular AliD:permease complex (**Fig 5B**). Our approach involved a systematic dissection of all interactions between AliD and the two permease subunits (AmiC and AmiD) (**S16 Fig**), revealing a complex interface characterized by numerous residues and interactions of diverse nature. Interestingly, when considering other OBP-permease complexes, the interaction involves the same regions of the OBPs and the permease, and most of interacting residues are conserved among different OBPs (**S16B Fig**).

The transmembrane domains (TMDs) of AmiC and AmiD, each consisting of six transmembrane helices, collectively form the channel responsible for facilitating the peptide passage through the lipid bilayer. Notably, AmiC is larger than AmiD (498 residues *vs* 308) and possesses a considerably larger extracellular portion. This particular region of AmiC (residues 31–278) is organized into two distinct extracellular domains (ECD1 and ECD2) (**Fig 5A**), which could play a relevant role in clamping the holo OBP. Specifically, the ECD1 is inserted as a wedge between both lobes of the OBP, while the ECD2 could act as an additional platform surface that facilitates further OBP recognition through interactions exclusively with Lobe 1 of the OBP (**Fig 5A**).

On the other hand, the extracellular part of AmiD primarily comprises a coil subdomain (residues 64–99, connecting TM1 and TM2 in AmiD) that interacts with Lobe 2 of the OBP.

The overall permease architecture is completed by two nucleotide-binding domains (NBDs), AmiE and AmiF located in the cytoplasm and equipped with all the necessary elements for ATP hydrolysis. Coupling helices, known to be responsible for dynamic transmission from NBDs to the TMDs [38], are also predicted for AmiC (residues 385–413) and AmiD (residues 197–219) (**Fig 5B**).

While AF offers an overall reliable 3D model, it lacks now the capability to predict the specific open and closed conformations of the Ami OBPs (**S15D Fig**), as well as the transition from the inward (closed) to the outward (open) conformation of the permease. Our structural analysis reveals that AF predictions for the permease in complex with the OBPs show an inward conformation, with a cluster of bulky aromatic residues (L435, L439 and I450 from AmiC; L155, F244, F247 and F248 from AliD) obstructing the passage (**Fig 5C** top), thereby impeding oligopeptide transport. Also, the NBDs are in their apo conformation (before ATP binding, **Fig 5B** top). To gain further insights into the structural transitions required for oligopeptide transport, we employed the recently developed program VARIO (http://chango.ibmb. csic.es/VAIRO), to provide functional context to AF-predicted models by incorporating consistent prior experimental knowledge (see Methods). This is necessary to guide AF predictions to less stable conformations, as in this case where energy provided by the ATPase assists conformational changes. Consequently, an outward conformation for the permease was achieved (**Fig 5B** and **5C**). This conformation reveals AmiE and AmiF in close contact, as observed in other NBDs from ABC transporters when trapping ATP [39] (**Fig 5B** bottom). In this conformation the transmembrane dimer formed by AmiC and AmiD adopts an outward open conformation, with displacement of the aforementioned cluster of hydrophobic residues, creating the space for oligopeptide transport (**Fig 5C** bottom). While a comprehensive description of the dynamics and molecular intricacies of oligopeptide transport awaits further experimental investigations, our study offers a thorough analysis of the initial step orchestrated by the OBP

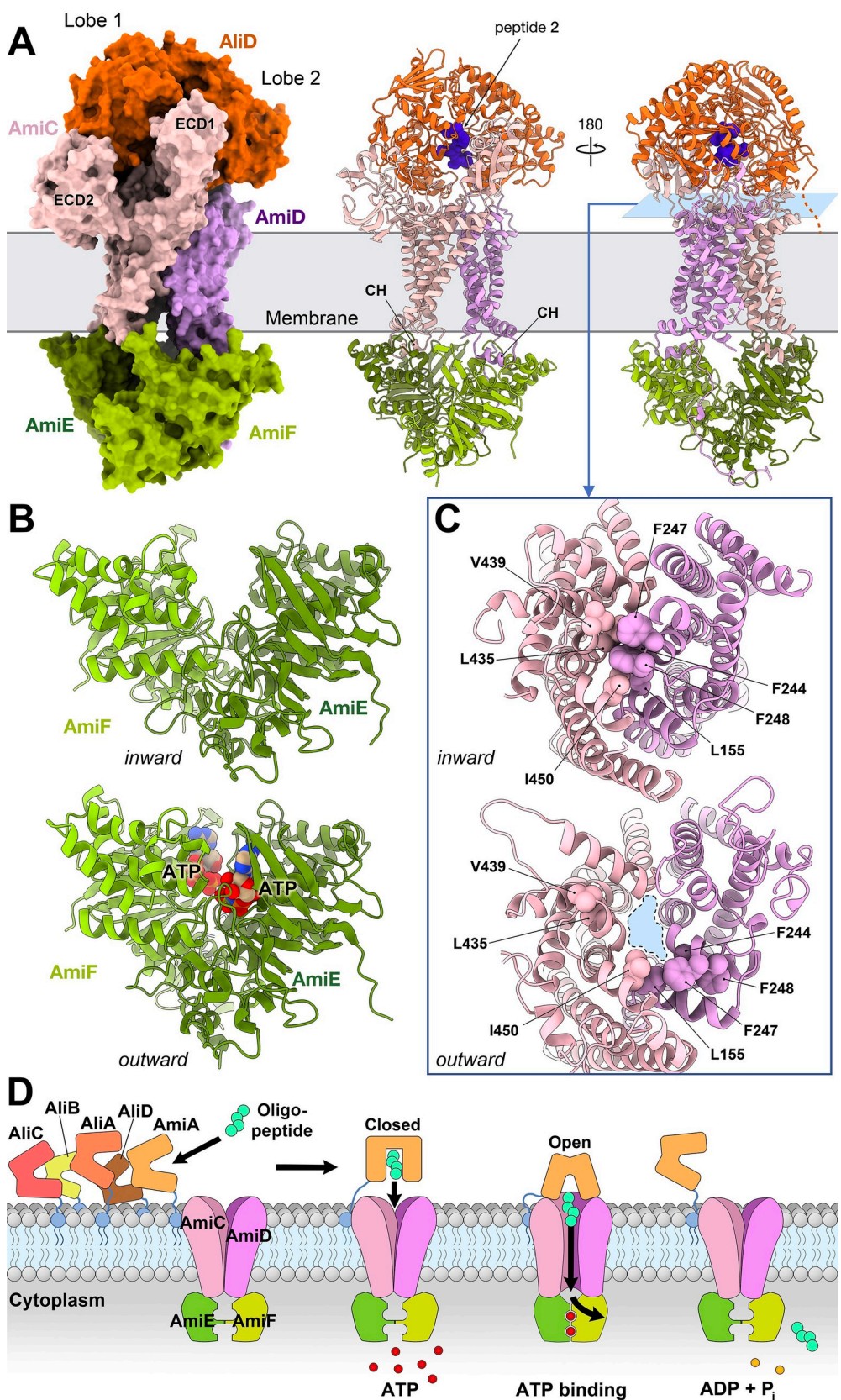

**Fig 5. AF Model for the oligopeptide transport system in *S. pneumoniae*.** (**A**) Left panel, surface representation of the AliD:permease complex predicted by AF. Each protein is color-coded and labeled accordingly. CH: coupling helix. (**B**) Different configurations of the NBDs in the inward (closed, upper panel) and outward (open, lower panel) conformations. AmiF and AmiD are represented in cartoon form, and the ATP molecules are shown as spheres. (**C**) View corresponding to a slice located at the blue plane depicted in A, right panel. The upper panel highlights the identified hydrophobic residues that obstruct the channel in the inward, closed conformation. In contrast, the lower panel shows the location of the same residues in the outward, open conformation. The generated cavity, presumably allowing oligopeptide transport, is marked in blue and outlined with a dashed black line. Relevant residues are labeled and represented as spheres. (**D**) Different proteins of the Ami system and conceptual schematic model for the mechanism of oligopeptide transport.

lipoproteins in the pneumococcal Ami system (**Fig 5D**). The importation of diverse oligopeptides allows pneumococcal growth despite several auxotrophies and provides crucial sensory input regarding local environment's composition. The additional OPBs, AliC and AliD, identified in some NESp strains, might enable the sensing of a broader range of bacterial competitors present in the company of these highly commensal organisms or incorporation of oligopeptides from pneumococcal non-competent siblings after the fratricide mechanism. A better understanding of how pneumococcus receives external input and responds to these stimuli is imperative to comprehend the transition from a commensal to a pathogenic state.

## Material and methods

### Recombinant DNA techniques, protein expression and purification

The construction of protein expression plasmids was carried out using standard protocols for PCR, molecular cloning, transformation, and DNA analyses. The coding sequences of AmiA, AliA, AliB, AliC and AliD (excluding their respective signal peptides) were PCR-amplified utilizing genomic DNA extracted from different pneumococcal strains (**S8 Table**) as a template. Gene-pecific oligonucleotides (**S9 Table**) and a high-fidelity DNA polymerase (Phusion; New England BioLabs) were employed in the amplification process. Gene fragments, once amplified, were inserted into the Isopropyl-β-D-1-thiogalactopyranoside (IPTG)-inducible expression vector pTP1 (**S8 Table**) through digestion/ligation cloning. All recombinant plasmids were verified by DNA sequencing. Subsequently, the sequenced plasmids were transformed into *E. coli* BL21 (DE3). Additionally, the AliD expression plasmid was transformed into Met-auxotroph *E. coli* 834 (DE3). For protein production in the cytoplasm of *E. coli* BL21 (DE3), the resulting recombinant strains harboring AmiA, AliA, AliC or AliD constructs, were cultured in LB medium supplemented with kanamycin (50 μg/ml) and grown to an $OD_{600}$ of 0.6–0.8 at 30°C. Protein expression was then induced with 1 mM IPTG during for 4 h.

The $His_6$-tagged proteins were purified by affinity chromatography using HisTrap HP Ni-NTA columns and the ÄKTA purifier liquid chromatography system (GE Healthcare) according to the instructions of the manufacturers. The $His_6$-tag was removed by TEV protease cleavage. The proteins (without $His_6$-tag) were dialyzed against 20 mM Tris-HCl (pH 8.0) and concentrated to 10–15 mg/ml using Vivaspin Ultra concentrators (Sartorius, Göttingen, Germany). Purity of the proteins was analyzed after sodium dodecyl sulfate polyacrylamide gel electrophoresis (SDS-PAGE) by both Coomassie brilliant-blue (CBB) staining and immunoblotting.

Expression of L-Selenomethionine (L-SeMet) labelled AliD protein was performed by growing a single colony of *E. coli* 834 (DE3) containing AliD expression construct overnight in 100 ml SelenoMet Medium (Molecular Dimensions, UK) supplemented with L-methionine. Bacteria were harvested, washed three times with 100 ml sterile water and resuspended in 6 ml sterile water. This culture was subsequently inoculated into 1 L SelenoMet Medium

prewarmed in 30˚C water bath and grown for 3 h at 30˚C. The expression of protein was induced by the addition of 1 mM IPTG and further growth was performed for 6 h at 30˚C. Bacteria were harvested by centrifugation. SeMet-labelled AliD was purified by affinity chromatography as described above.

## Oligopeptides synthesis

**Instrumentation and Chemicals.**   Unless otherwise specified, analytical grade solvents and commercially available reagents were used without further purification. DIEA, piperidine, Ac$_2$O and TFA were purchased from Aldrich (Germany), and O-(6-chlorobenzotriazol-1-yl)-1,1,3,3-tetramethyluronium hexafluorophosphate (HCTU) from Fluorochem (UK). Fmoc-protected Rink Amide MBHA resin (0.71 mmoL/g loading) was purchased from Iris Biotech (Germany). Fmoc-α-protected amino acids were purchased from Fluka (Germany), Novabiochem (Merck, Germany) and Iris Biotech (Germany). All aminoacids used were of the L configuration.

All peptides were synthesized manually on resin following the standard Fmoc/tBu solid-phase orthogonal protection strategy on a 20-position vacuum manifold (Omega, Bienne, Switzerland) connected to a vacuum pump using 20-mL polypropylene plastic syringes (Dubelcco) with a preinserted frit and a Teflon stopcock to do the washings and remove the solvents and excess reagents. The coupling reactions were carried out on solid phase using an IKA-100 orbital shaker.

The final peptides were purified on a semipreparative HPLC Waters equipment. As mobile phase, mixtures of A/B were used, where A = 0.05% TFA water and B = acetonitrile with a flow rate of 7 mL/min, using a gradient from 0% of B to 100% of B in 30–45 min and were detected at 217 nm. After purification, peptides were lyophilized (using water / acetonitrile mixtures on a Telstar6-80 instrument) and dried under reduced pressure in the presence of P$_2$O$_5$.

The purity of the final products was checked by analytical RP-HPLC on an Agilent Infinity instrument equipped with a Diode Array and a C18 Sunfire column (4.6 mm × 150 mm, 3.5 μm). As mobile phase, mixtures of A/B were used, where A = 0.05% TFA water and B = acetonitrile with a flow rate of 1 mL/min. The peptides were analyzed at 217 and 254 nm in a gradient from 10% of B to 100% of B in 10 min. The HPLC chromatograms indicated more than 97% purity. HRMS (EI+) was carried out in an Agilent 6520 Accurate-Mass Q-TOF LC/MS spectrometer using water/acetonitrile.

## Solid-phase peptide synthesis (SPPS) General Protocol

Fmoc-protected Rink Amide MBHA resin was swollen in DCM/DMF/DCM/DMF (3 × 0.5 min), in a filter-equipped syringe. Then, the resin was treated with 20% piperidine in DMF at room temperature (1 × 1 min and 3 × 10 min) and washed with DMF/DCM/DMF/DCM (3 × 0.5 min, each solvent). Next, to the free Nα-terminal swollen resin (1 equivalents), a solution of the corresponding Fmoc-α-amino acid (1.2 equivalents), HCTU (2 equivalents) and DIEA (4 equivalents) in dry DMF (5 mL) was added. The mixture was stirred at room temperature for 1h. Finally, the resin was drained and washed extensively under vacuo (DMF/DCM/DMF/DCM, 5 × 0.5 min) and dried under vacuum. This protocol was repeated for the sequential anchoring of each amino acid until sequence completion. Coupling reactions to primary amines were monitored by the Kaiser ninhydrin test, and to secondary amines by a more sensitive Choranil test. Half-way through the growing sequence, the progress of the reactions was followed by the HPLC-MS analysis of a small sample of peptidyl resin after acidic cleavage.

Once the sequence of the peptides was completed, these were cleaved from resin by treatment with TFA/TIPS/H$_2$O 95:2.5:2.5 (5 vol) for 4 h at room temperature. The filtrates were

precipitated over cold Et$_2$O, in a and centrifuged three times at 5000 rpm for 10 min on a Hettlich Universal 320R centrifuge (Sigma-Aldrich). The supernatant was removed, and the process was repeated three times. After supernatant removal, the pellet was suspended in a mixture of in water/acetonitrile and lyophilized. Finally, the crudes were purified using a preparative reverse phase HPLC Waters equipment connected to a Fraction Collector III, using a C18 ACE 5 C18-300 (250 × 10 mm) column. The mobile phases consisted on acetonitrile / water / 0.1% of formic acid as in a gradient of 2–95% of acetonitrile in 30 min with a flow of 6 mL/min. Samples were loaded dissolved in the minimal quantity of water/acetonitrile/DMSO. The peptides were detected at 217 nm.

**Peptide 1** (FPPQSV). The general protocol was followed starting from 0.14 mmol of resin. The final residue was purified to yield Peptide 1 as a white lyophilized cotton-like solid (17 mg, 18% overall yield). Analytical HPLC (gradient 10–100% of acetonitrile in 10 min), retention time = 4.27 min (99% analytical purity, at 217 nm). HRMS (ESI,+) m/z was calculated for C$_{32}$H$_{48}$N$_8$O$_8$ 672.3614 and found [M+H]$^+$ 673.3697, [M+Na]$^+$ 695.3501 (2.85 ppm).

**Peptide 2** (AIQSEKARKHN). Starting from 0.14 mmol of resin, and following the general protocol Peptide 2 was purified and isolated as a white lyophilized cotton-like solid (35 mg, 19% overall yield). Analytical HPLC (gradient 10–100% of acetonitrile in 10 min), retention time = 3.94 min (100% analytical purity, at 217 nm). HRMS (ESI,+) m/z was calculated for C$_{53}$H$_{93}$N$_{21}$O$_{16}$ 1279.70999 and found [M+H]$^+$ 1280.71788 (0.72 ppm).

**Peptide 3** (PIVGGHEGAGV). The general protocol was followed with 0.14 mmol of resin. After purification of the crude, Peptide 3 was isolated as a white lyophilized cotton-like solid (29 mg, 21% overall yield). Analytical HPLC (gradient 10–100% of acetonitrile in 10 min), retention time = 3.95 min (98% analytical purity, at 217 nm). HRMS (ESI,+) m/z was calculated for C$_{43}$H$_{70}$N$_{14}$O$_{13}$ 990.5265 and found [M+H]$^+$ 991.5338 (1.84 ppm).

**Peptide 4** (VMVKGPGPGREST). The general protocol was followed with 0.14 mmol of resin. After purification of the crude, Peptide 4 was isolated as a white lyophilized cotton-like solid (26 mg, 14% overall yield). Analytical HPLC (gradient 10–100% of acetonitrile in 10 min), retention time = 3.65 min (100% analytical purity, at 217 nm). HRMS (ESI,+) m/z was calculated for C$_{55}$H$_{96}$N$_{18}$O$_{17}$S 1312.6917and found [M]$^+$ 1312.6922 (0.33 ppm).

**Peptide 5** (AKTIKITQTR). Starting from 0.14 mmol of resin, and after following the general protocol Peptide 5 was purified and isolated as a white lyophilized cotton-like solid (26 mg, 16% overall yield). Analytical HPLC (gradient 10–100% of acetonitrile in 10 min), retention time = 3.71 min (97% analytical purity, at 217 nm). HRMS (ESI,+) m/z was calculated for C$_{50}$H$_{95}$N$_{17}$O$_{14}$ 1157.7279 and found [M+H]$^+$ 1158.7350 (2.96 ppm).

## Protein crystallization

After purification, AliD, AliB, AliC and AmiA were concentrated up to 17, 18, 25 and 30 mg/ml; respectively, using a Millipore ultra-concentrator (10 kDa cutoff). Crystallization screenings were performed by high-throughput techniques in a NanoDrop robot and Innovadyne SD-2 microplates (Innovadyne Technologies Inc.), screening PACT Suite and JCSG Suite (Qiagen), JBScreen Classic 1–4 and 6 (Jena Bioscience) and Crystal Screen, Crystal Screen 2 and Index HT (Hampton Research). The conditions that produced crystals were optimized by sitting-drop vapor-diffusion method at 291 K by mixing 1 μL of protein solution and 1 μL of precipitant solution, equilibrated against 150 μL of precipitant solution in the reservoir chamber. The best crystals for each protein were obtained in the conditions indicated below. AliD in an opened conformation crystallized in 0.2 M MgCl$_2$, 23% PEG3350 and 0.1 M Tris-HCl pH = 8.0. AliD:peptide **1** complex crystallized in 150 mM Zn acetate, 10% PEG8000, 0.1 M MES pH = 6.5. AliB:peptide **2**, AliB:peptide **3** and AliB: peptide **4** complexes were crystallized

in 0.1M sodium acetate, 8% 2-Propanol and 21% PEG4000. AliC in an opened conformation crystallized in 0.1 M Bis-Tris pH = 6.5 and 25% PEG3350. AmiA:peptide **5** complex crystallized in 0.1 M citric acid and 25% PEG3350.

**Co-crystallization experiments with oligopeptides.** For co-crystallization trials, all peptides were dissolved in water up to 50 mM and incubated with the corresponding protein at a final concentration ranging 3–5 mM for 30 min at 18˚C (unless indicated) using the crystallization conditions described above.

## Data collection and structural determination

In all cases, crystals were cryo-protected in the precipitant solution supplemented with 30% (v/v) glycerol, mounted into nylon loops and flash cooled in liquid nitrogen at 100 K. Diffraction data was collected in beamline XALOC at the ALBA synchrotron (CELLS-ALBA, Spain), using a Pilatus 6M detector. AliD structure determination was performed using a SeMet derivative in order to solve the phase problem associated with the structure determination of native AliD. Native and SAD experiments were carried out and the collected datasets were processed with XDS [40] and Aimless [41]. AmiA was solved ab initio by using the Arcimboldo program SHREDDER [42,43] deriving fragments from a template with 26% sequence identity. *SEQUENCE SLIDER* [44] was used to extend starting partial polyalanine models with side chains in plausible ways to increase the scattering and allow successful extension with SHELXE [45]. AliD structure was solved by SAD technique. The protein has 6 Methionine residues. The initial selenium sites were located with SHELX [45,46] using the HKL2MAP GUI [47]. Once the AliD structure was solved, this model was used as a template in the molecular-replacement method using MOLREP [48] and Phaser [49] to solve the remaining structures. Refinement and manual model building of all structures was performed with Phenix [50] and Coot [51], respectively. The stereochemistry of the final model was checked by MolProbity [52]. In all cases refinement strategy included atomic coordinates, individual B-factors, TLS parameters, occupancies and automatic correction of N/Q/H errors. Data collection and processing statistics are shown in **S10 Table**.

## Identification of peptides bound to AmiA and AliB by Mass Spectrometry

Peptides bound to AliB and AmiA during their expression in *E. coli*, were isolated as previously described [9]. ZipTips (Merck KGaA, Darmstadt, Germany) were used to desalt the peptide containing samples prior MS-analysis. Therefore, the samples were loaded from the top onto the C18 material and washed according to the manufacturer protocol. Elution of the samples was done in two steps, starting with 50% ACN followed by 80% ACN. Both eluates were combined and freeze-dried before reconstitution in buffer A (0.5% DMSO in water with 0.1% acetic acid). LC separation and MS measurement were performed using a NanoAcquity UPLC (Waters GmbH, Eschborn, Germany) and a LTQ-Orbitrap Velos mass spectrometer (Thermo Fisher Scientific, Dreieich, Germany), details are given in **S11 Table**.

The obtained data were searched using the SEQUEST search engine (Thermo Fisher Scientific, San Jose CA, USA; version 1.0), implemented in the Sorcerer XT (Sage-N Research, Milpitas CA, USA) using *E. coli* BL21 database assuming non enzyme digestion, methionine oxidation as variable modification, an ion mass tolerance of 1.00 Da and a parent ion tolerance of 10 ppm. Scaffold (version Scaffold_5.3.0, Proteome Software Inc., Portland, OR) was used to validate MS/MS based peptide identifications. Peptide identifications were accepted if they could be established at greater than 99.0% probability by the Scaffold Local FDR algorithm [53]. Peptide candidate lists were exported to Microsoft Excel (Microsoft, Redmond, USA) for further analysis.

### Prediction of 3D structures guiding AF with prior information

The program VAIRO guides AF predictions towards a particular functional state selecting consistent prior experimental knowledge relevant to that state and masking or down weighting information that would otherwise dominate. In this work, all subunits forming the permease system and the five different OBPs have been predicted alone and in complexes in different conformational states. Using as templates the experimental structures of SBPs in open, unbound state or alternatively, those in closed, ligand-bound state led to matching conformations for the AF predictions, yielding models representing states for which no experimental structure was available. High pLDDT scores and favorable free energy characterized these predicted models. Furthermore, predictions corresponding to available experimental structures have been compared to assess the reliability of the method, evidencing deviations from the experimental structure below 1.1 Å *rmsd*. AliA, which is not available experimentally in either state, was predicted in open and closed conformations. The TMD formed by AmiC and AmiD was guided towards an outward conformation promoting the extracellular/periplasmic aperture of the channel using as a template the transmembrane domain of a maltose transporter in pre-transitional state [3PUY [54]]. The resulting models display the hydrophobic residues that are blocking the channel displaced towards the external region of the channel. The NBD formed by AmiE and AmiF was guided towards its catalytic conformation using as a template the NBD of the same experimental structure used to predict the TMD in outward facing. The prediction obtained shows AmiE and AmiF closer, being compatible with a conformation responsible for the hydrolysis of the ATP. Finally, predictions of the full permease system including as templates the previous predictions for all different subunits in their respective outward and catalytic conformation was made.

### Microscale thermophoresis assays

Binding of peptides to heterologously expressed AliD was measured by microscale thermophoresis (MST). Therefore, AliD was fluorescent-labeled using the labeling kit RED-NHS (2nd generation, NanoTemper Technologies). The labeling procedure was performed according to the manufacturer's instructions. Briefly, phosphate buffered saline (PBS, pH 7.4) was used as labeling buffer and 10μM AliD was labeled for 30 minutes in the darkness. Unbound dye was removed by size exclusion chromatography using the supplied column B. Finally, the labeled AliD was eluted with PBS in a final concentration of 2μM. Analyzed peptides (1: FPPQSV and 2: AIQSEKARKHN) were diluted in PBS. For the binding experiments, different dilutions of the peptides ($7.63 \times 10^{-6}$–0.125 mM) were prepared in PBS and mixed with 20 nM labeled AliD. Samples were loaded into standard capillaries (NanoTemper Technologies) and measured using the Monolith NT.115 machine (NanoTemper Technologies). All measurements were performed at 25˚C, 40% LED power and medium MST power. Data of three independently conducted experiments were analyzed using the Mo.Affinity Analysis software version 2.3 from NanoTemper Technologies.

### Analytical ultracentrifugation assays

Experiments were performed in an Optima XL-I analytical ultracentrifuge (Beckman-Coulter Inc.) equipped with both UV-VIS absorbance and Raleigh interference detection systems, using an An-50Ti rotor and 12 mm optical pass epon-charcoal standard double sector centerpieces. Samples of AliD and AliC in 20 mM Tris–HCl (pH 7.5) were centrifuged at 48,000 at 20˚C. Differential sedimentation coefficient distributions were calculated by least-squares boundary modelling of sedimentation velocity data using the continuous distribution c(s) Lamm equation model as implemented by SEDFIT software.

## Supporting information

**S1 Table. Percent Identity Matrix created by Clustal2.1 [1].**
(DOCX)

**S2 Table. Chemically synthesized oligopeptides used in this work along with their corresponding Oligopeptide-binding protein (OBP). Table adapted from Fauzy Nasher *et al*; 2018 [2].**
(DOCX)

**S3 Table. Detailed composition for each substrate-binding pocket and its physicochemical properties in the AliD:peptide 1 complex.**
(DOCX)

**S4 Table. Detailed composition for each substrate-binding pocket in AliB:peptide 2 complex.**
(DOCX)

**S5 Table. Detailed composition for each substrate-binding pocket in AliB:peptide 3 complex.**
(DOCX)

**S6 Table. Detailed composition for each substrate-binding pocket in AliB:peptide 4 complex.**
(DOCX)

**S7 Table. Detailed composition for each substrate-binding pocket in AmiA:peptide 5 complex.**
(DOCX)

**S8 Table. Strains and plasmids list.**
(DOCX)

**S9 Table. Primer list.**
(DOCX)

**S10 Table. Crystallographic data collection and refinement statistics.**
(DOCX)

**S11 Table. Nano LC-MS/MS data acquisition parameters.**
(DOCX)

**S1 Fig. Sequence comparison among the OBPs of the Ami system.** Sequence alignment produced by T-COFFEE [22] and drawn with ESPript [23]. Identities are boxed in black. Similarities are boxed in gray according to physicochemical properties. Secondary structure elements have been calculated from the 3D structure of AliD (this work) using the program DSSP [24]. They are displayed on the top of sequence blocks. Alpha and $3_{10}$ helices are represented by squiggles labelled α and η, respectively. Strands are represented by arrows. Sequence numbering corresponds to AliD sequence. Regions R1 and R2 are highlighted in green and salmon, respectively (see main text for details). Consensus: a consensus sequence is generated using criteria from MultAlin: uppercase is identity, **l**owercase is consensus level > 0.5,**!** is anyone of IV, **$** is anyone of LM, **%** is anyone of FY, **#** is anyone of NDQEBZ. Lowercase is consensus level > SimilarityGlobalScore if S, M or E are used as SimilarityType. See t-Coffe similarity score description for further details.
(TIF)

**S2 Fig. Electron-density map (2mFo-DFc map contoured at 1.0 σ) of the AliD structure at a resolution of 1.8 Å in the open conformation.** The protein structure is depicted in blue capped sticks. A $Mg^{2+}$ atom, contributing to crystal packing, is represented as a green sphere. The lower panel provides a detailed view of the boxed area in the upper panel, highlighting electron densities at 1.8 Å resolution.
(TIF)

**S3 Fig. Structural and sequence comparison of AliD and OppA from S. typhimurium.** (**A**) Left panel; superposition of the structures of AliD from *S. pneumoniae* (this work, colored in red) and OppA from *S. typhimurium* [colored in gray, PDB 1RKM [25]], both in the open conformation (*rmsd* of 3.18 Å for 349 Cα atoms). Right panel; structure superposition between AliD from *S. pneumoniae* (this work, colored in red) and OppA from *S. typhimurium* [colored in gray, PDB 1B4Z [26]], both in the closed conformation (*rmsd* of 3.20 Å for 347 Cα atoms). Both proteins are represented as cartoon oval. The two crossover connecting regions in AliD are colored yellow and labeled as "c1" and "c2", respectively. (**B**) Sequence alignment of AliD from *S. pneumoniae* and OppA from *S. typhimurium*, generated by T-COFFEE [22] and drawn with ESPript [23]. Sequence identity, estimated with Clustal2.1, is 25,76%. Identities are boxed in black. Similarities are boxed in gray according to physicochemical properties.
(TIF)

**S4 Fig. Positively-charged residues on the C-terminal α-helix and Lobe 1 of AliD.** Left panel, cartoon representation of the AliD structure (open conformation) using the same color code as Fig 2B. The focus is on the C-terminal α-helix (α19), where positively-charged residues (mainly lysines) are depicted as capped sticks. Right panel, the electrostatic-potential surface of AliD is shown in the same orientation as the left panel. Relevant basic patches are indicated with black arrows. The color key (blue, positive and red, negative) shows the Poisson-Boltzmann electrostatic-potential surface (color bar range ± 68.99 kT/e).
(TIF)

**S5 Fig. Domain-swapped structure of AliC in an open conformation.** (**A**) The crystal structure of the AliC dimer is depicted in ribbon form, with each monomer colored in green and orange, respectively. The domain swap in AliC results in an extended interface area of 5173.1 $Å^2$ [calculated with PDBe PISA v1.52 [28]], occurring at a hinge axis passing through the middle of the loop connecting α7 with β14 (residues 290–294), extending between Lobe 1 and Lobe 2. This loop in AliC contains a glycine residue (G290) that is unique compared to other homologs, which typically have a lysine residue at that position (K287 in AliD, see **S1 Fig**). The electron density for the hinge loop (boxed inset) clearly defines the connection between the lobes of each monomer in the AliC dimer, with the residues shown as capped sticks. (**B**) Structural superposition of AliD (colored in gray) with one of the domain-swapped monomers of AliC (colored in green and orange). The width of the peptide-binding cavity for both proteins is indicated with arrows (16 Å in AliD *vs* 29 Å in AliC). N, amino-terminus; C, C-terminus. (**C**) Sedimentation coefficient distributions [c(s)] of AliD (black line) and AliC (red line) at 4.1 μM. Note that the s-values shown have not been corrected for buffer composition.
(TIFF)

**S6 Fig. Peptide 1 structure in the AliD:peptide 1 complex.** (**A**) Binding of peptide **1** and **2** to heterologously expressed AliD as analyzed by microscale thermophoresis (MST). The concentration of NHS-RED-labeled AliD was kept constant (20 nM), while the concentration of the non-labeled peptides ranged between $7.63x10^{-6}$–0.125 mM. Samples were measured using the Monolith NT.115 (NanoTemper Technologies) at 40% LED power and medium MST power at 25° C. The Kd was calculated from three independent measurements, error bars represent

the standard deviation. (**B**) Structural superposition of the two peptide **1** molecules (depicted as capped sticks) bound to AliD monomers A (gray) and B (yellow). The sequence of peptide **1** (FPPQSV) is indicated and numbered. N-term, amino-terminus; C-term, C-terminus. (**C**) Atomic B factors for peptide **1** as observed in the AliD:peptide **1** complex (monomer B). The Ligand is represented as capped sticks and colored based on the B-factor distribution, ranging from low (blue) to high (red) values. (**D**) Electron-density map (2mFo-DFc map contoured at 1.0 σ) for each of the two peptide **1** molecules (in green caped sticks) observed in the AliD:peptide **1** complex. The position of each peptide residue is numbered.
(TIF)

**S7 Fig. Representation of mFo-DFc maps (contoured at 2.5 σ, red mesh) corresponding to the ligands within all presented complexes.** The 2mFo-DFc maps (contoured at 1.0 σ, blue mesh) for each peptide, as acquired post-refinement are also displayed. Peptides are depicted in green capped sticks.
(TIF)

**S8 Fig. An unidentified oligopeptide was discovered bound to AliB during the expression and purification of the AliB protein in E. coli.** (**A**) Cartoon representation of AliB structure (colored green) in complex with an unknown oligopeptide(s) from *E. coli* modeled as poly-Ala (depicted as yellow capped sticks). (**B**) Electron-density map (2mFo-DFc map contoured at 1.0 σ) of the unknown ligand in which a nine-residues-long poly-alanine backbone has been traced (depicted as yellow caped sticks). The ligand is presented in a similar orientation to that in panel A. The positions for each alanine residue are indicated. N-term, amino-terminus; C-term, C-terminus.
(TIF)

**S9 Fig. Structural superposition between AliB and AliD in the closed conformation.** Lower panel, the overall closed conformations of AliD (colored in salmon) and AliB (colored in pale purple) exhibit significant similarity (*rmsd* 0.88Å for 588 Cα atoms superposition), although AliB's structure is more tightly closed compared to AliD. Both proteins are depicted in ribbon representation. The upper panel provides a closer view of the region boxed in the lower panel. Differences in the loops shaping the substrate-binding site at positions 1–4 are highlighted with arrows (see main text for details). Peptide **1** (FPPQSV) is depicted as yellow-capped sticks.
(TIF)

**S10 Fig. Substrate trapped in AmiA during the purification process.** (**A**) Ribbon representation of AmiA structure in complex with an unknown peptide from *E. coli* (in yellow caped sticks) that we further refined as peptide **4**. (**B**) Electron-density map (2mFo-DFc map contoured at 1.0 σ) for the 10-residues long ligand has been traced (yellow caped sticks) assuming the sequence of peptide **4** (AKTIKITQTR). Ligand is presented in a similar orientation as the one found on panel A. Positions for each residue are indicated.
(TIF)

**S11 Fig. Electron density maps for peptide 5.** (**A**) 2mFo-DFc electron-density map for peptide **5** (AKTIKITQTR depicted in green caped sticks) for each of the four monomers (monomers A, B, C and D) as observed in the crystallized AmiA:peptide **5** complex. (**B**) Atomic B factors for peptide **5** (monomer C). The Ligand is represented as capped sticks and colored according to the B factor distribution, ranging from low (blue) to high (red) values. (**C**) Structural superposition among the peptide **5** molecules (showed in capped sticks) as observed in the AmiA complexes. Lateral chains from AmiA residues that adopt different conformations

are labeled. The peptide sequence is indicated and numbered. N-term, amino-terminus; C-term, Carboxy-terminus. Maps contoured at 1.0 σ.
(TIF)

**S12 Fig. LC-MS identification of oligopeptides bound to AmiA and AliB during production and purification.** (**A**) The plots show the frequency distribution of peptide lengths that are found bound to AmiA (left panel) or AliB (right panel). The data are represented as mean ± SE calculated from non-linear Gaussian fit version (full width at half maximum, FWHM) employing the function:

$$y = y0 + \frac{Ae^{\frac{-4ln(2)(x-xc)^2}{w^2}}}{w\sqrt{\frac{\pi}{4ln(2)}}}$$

(**B**) Sequence logo analysis of the peptides bound to AmiA (left panel) or AliB (right panel). The sequence logos were generated based on the alignment of the identified peptides bound to each respective protein, taking just into account their starting positions for the alignment. These plots were generated using the WebLogo server (https://weblogo.berkeley.edu/logo.cgi) [29]. (**C**) Sector graphs comparing the identified bound peptides to AmiA (shown in the upper panel) and AliB (shown in the lower panel). The peptides were categorized according to the biological function exerted by the protein from which they originated.
(TIF)

**S13 Fig. Substrate-binding cavities in OBPs of the Ami system and comparison with related proteins.** (**A**) The bound peptides visualized inside the binding cavity of OBPs belonging to the Ami permease reveal the extent of the unoccupied space. The cavity is shown as a gray semi-transparent surface, the peptide is shown in yellow capped stick, and the protein is not shown. AmiA:**5**, AmiA in complex with peptide **5**; AliD:**1**, AliD in complex with peptide **1**; AliB:**2**, AliB in complex with peptide **2**; AmiB:**3**, AmiB in complex with peptide **3** and AmiB:**4**, AmiB in complex with peptide **4**. (**B**) Comparison of the binding cavity volumes found on SBPs of the Ami permease, including the cavities of OppA from *L. lactis* [PDB 3DRF [30]], AppA from *B. subtillis* (PDB 1XOC [31]), OppA from *S. typhimurium* [PDB 1B4Z [26]], DppA [PDB 1DPE] and ProX from *E. coli* [PDB 1SW2 [33]], for comparative purposes. The volumes were calculated with the program POCASA v1.1 [34] using default parameters and a Probe Radius of 1 Å.
(TIF)

**S14 Fig. Binding sites in OBPs of the Ami system.** (**A**) Left, Structural superimposition of Lobe 2 from AliB: peptide **2**, AliB:peptide **3** and AliB:peptide **4** complexes. Right, Electrostatic potential surface of Lobe 2 AliB:peptide **2** complex showing peptide recognition and binding pockets (see main text). (**B**) Electrostatic potential surface of Lobe 2 in AliD:peptide**1** complex, AliC, AmiA:peptide **5** complex and in AliA as predicted by AlphaFold (AF). P1 to P4, pocket 1 to pocket 4.
(TIF)

**S15 Fig. AF model for the complete Ami transporter system.** (**A**) Cartoon representation of the predicted AF structure of the full-length Ami permease in complex with AliD (color-coded by model confidence). (**B**) Cartoon representation of the predicted AF structure of the full-length AliD (color-coded by model confidence). (**C**) Structural superposition between the AliD structure determined by X-ray crystallography (colored in salmon) and the structure predicted by AF (colored in gray, *rmsd* of 0.864 Å across 591 pruned atom pairs). Both structures

are displayed in cartoon representation. (**D**) A structural superposition was carried out among 10 AliD structures predicted by AF. The *rmsd* values ranged from 0.07 to 0.53 Å across 601 pruned atom pairs. All structures were aligned with respect to lobe 2 and are depicted in ribbon representation, each represented by a distinct color.
(TIFF)

**S16 Fig. Residues of AliD interacting with the permease in the AF model.** AliD residues interacting with the two permease subunits (AmiC and AmiD) were identified with LigPlot$^{+}$ v.2.2 [35]. (**A**) Upper panel, cartoon depiction of AliD (colored orange) is presented in a manner akin to that in Fig 2B (upper panel). AliD residues involved in the interaction with the permease are labeled and depicted in yellow capped sticks, while the semi-transparent representation illustrates the AliD surface. Peptide **1** is visualized as blue spheres. The boxed inset exhibits a surface representation of AliD (orange), highlighting the positions of residues engaging with AmiC (pink) and AmiD (purple). The orientation of AliD corresponds to that in A, upper panel. The middle panel mirrors the representation in A, upper panel, but with a 60˚ rotation through the indicated axes. The lower panel replicates the view from A, middle panel, incorporating the two permease subunits, AmiC (pink cartoon), and AmiD (soft purple cartoon). ECD, extra-cellular domain. (**B**) In the upper panel, a table is provided summarizing AliD residues involved in interactions with AmiD. The lower panel presents a table summarizing AliD residues engaged with AmiC. The symbol '*' denotes residues that are absolutely conserved across all five OBPs of the Ami system. The symbol '#', on the other hand, designates residues with physicochemical properties conserved throughout all five OBPs of the Ami system.
(PDF)

**S1 Movie. Movie illustrating the Venus fly-trap mechanism for the particular case of AliD complexed with peptide 1.** The structure of AliD is presented in a cartoon oval representation. Domain I, II and III are color-coded as green, pale brown and blue, respectively. Peptide **1** is represented as yellow capped sticks. Hydrophobic residues, forming a hydrophobic patch that encloses the ligand when the protein adopts a closed conformation, are also visualized in capped sticks.
(MPG)

## Acknowledgments

We are grateful to the staff of XALOC beam line at ALBA synchrotron facility (Barcelona, Spain) for support in diffraction data collection. We thank Vishnu Mukund Dhople at the Department for Functional Genomics (University Medicine, Greifswald, Germany) for the recording of the mass spectrometry data.

## Author Contributions

**Conceptualization:** Martín Alcorlo, Isabel Usón, Lance E. Keller, Jessica L. Bradshaw, Larry S. McDaniel, Uwe Völker, Sven Hammerschmidt, Juan A. Hermoso.

**Data curation:** Martín Alcorlo, Mohammed R. Abdullah, Thomas P. Kohler.

**Formal analysis:** Martín Alcorlo, Leif Steil, Larry S. McDaniel, Juan A. Hermoso.

**Funding acquisition:** Sven Hammerschmidt, Juan A. Hermoso.

**Investigation:** Martín Alcorlo, Mohammed R. Abdullah, Leif Steil, Francisco Sotomayor, Laura López-de Oro, Sonia de Castro, Sonsoles Velázquez, Thomas P. Kohler, Elisabet

Jiménez, Ana Medina, Isabel Usón, María-José Camarasa, Sven Hammerschmidt, Juan A. Hermoso.

**Methodology:** Leif Steil, Sonsoles Velázquez, Thomas P. Kohler, Elisabet Jiménez, Ana Medina, Isabel Usón, María-José Camarasa, Sven Hammerschmidt.

**Project administration:** Sven Hammerschmidt, Juan A. Hermoso.

**Software:** Elisabet Jiménez, Ana Medina, Isabel Usón.

**Supervision:** Sven Hammerschmidt, Juan A. Hermoso.

**Validation:** Martín Alcorlo, Uwe Völker.

**Writing – original draft:** Martín Alcorlo, Sven Hammerschmidt, Juan A. Hermoso.

**Writing – review & editing:** Martín Alcorlo, Lance E. Keller, Jessica L. Bradshaw, Larry S. McDaniel, María-José Camarasa, Uwe Völker, Sven Hammerschmidt, Juan A. Hermoso.

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
