## [Decision Letter · Decision Letter 0]

8 Jan 2024

Dear Prof. Hermoso,

Thank you very much for submitting your manuscript "Molecular and Structural Basis of Oligopeptide Recognition and Transport by the Ami Transporter System in Pneumococci" for consideration at PLOS Pathogens. As with all papers reviewed by the journal, your manuscript was reviewed by members of the editorial board and by several independent reviewers. In light of the reviews (below this email), we would like to invite the resubmission of a significantly-revised version that takes into account the reviewers' comments.

We cannot make any decision about publication until we have seen the revised manuscript and your response to the reviewers' comments. Your revised manuscript is also likely to be sent to reviewers for further evaluation.

Sincerely,

Gongyi Zhang

Academic Editor

PLOS Pathogens

Michael Otto

Section Editor

PLOS Pathogens

Kasturi Haldar

Editor-in-Chief

PLOS Pathogens

orcid.org/0000-0001-5065-158X

Michael Malim

Editor-in-Chief

PLOS Pathogens

orcid.org/0000-0002-7699-2064

Reviewer's Responses to Questions

**Part I - Summary**

Reviewer #1: Pneumococcus (Streptococcus pneumoniae) is a leading cause of pneumonia and is auxotrophic for cysteine, glycine, valine, leucine, isoleucine, arginine, asparagine, histidine, and glutamine. The uptake of diverse oligopeptides enables pneumococcal growth despite auxotrophies and functions as a critical sensor for assessing the composition of the local environment. The uptake of oligopeptides by pneumococci is performed by a specific ABC transporter system known as ‘Ami’. The Ami system includes two transmembrane proteins, AmiC and AmiD, two ATPases, AmiE and AmiF, and five oligopeptide-binding lipoproteins (OBPs), AmiA, AliA, AliB, AliC and AliD. This study presents nine crystal structures of four of the five OBPs within the Ami system, AmiA, AliB, AliC and AliD in two different conformations, closed (with peptide bound) and open. In addition, the authors also performed mass spectrometry analysis to identify the oligopeptides captured by AliB and AmiA during purification. Overall, this study provides valuable information about peptide transport mediated by OBPs within the pneumococcal Ami permease system. However, the manuscript suffers from two aspects and are unsuitable for publication at its current state.

Reviewer #2: In this manuscript, Alcorlo. Et. Al. report crystal structures of all the Oligopeptide Binding Proteins (OBPs) of the pneumococcal Ami ABC-transporter system. The authors show the detailed molecular mechanism of how OBPs interacts with oligopeptides. However, the authors do not properly show any biochemical assays to support what their structural observations, and do not report the final refinement validation files of their structures. More importantly, in the manuscript, the authors describe primarily oligopeptide recognition of OBPs, does not provide any experimental evidence for model of oligopeptide transport although the AliD : permease complex predicted by AF, which does not match to their title. Collectively, the manuscript cannot be published until the proper biochemical data are supplemented.

Reviewer #3: This manuscript by Alcorlo et al focuses on the structure of Ami/Ali proteins from Streptococcus pneumoniae. These proteins are substrate-binding proteins of ABC transporters that import oligopeptides into the cytoplasm. Because S. pneumoniae is an auxotroph for several amino acids, this ABC system is very important during infection by allowing the acquisition of peptides that can serve as sources of these amino acids. This manuscript describes the structure of several oligopeptide-binding Ami/Ali proteins in their apo and substrate-bound states. Some of the peptides bound to Ami/Ali proteins were synthetic peptides already known to bind to these proteins, while others were trapped during protein production in E. coli. From these data on substrate specificity, the authors classified the paralogous Ami/Ali proteins into two groups: the AliB-like group and AmiA-like group. Lastly, using modeling, the authors propose a model for the structure and function of this oligopeptide ABC transport system.

Overall, this is a beautiful structural study on a peptide transport system that is important in the physiology of the human pathogen S. pneumoniae. The manuscript is very well written, the data are of very high quality, and the figures are highly effective. The model in Fig. 5 is supported by the data and makes sense based on what we already know about other ABC transporters. However, as detailed below, the study needs some additional data to validate the authors’ conclusions on substrate specificity.

**Part II – Major Issues: Key Experiments Required for Acceptance**

Reviewer #1: Key points that need to be addressed are:

1. The Ami system has five OBPs for one ABC transporter, AmiA, AliA, AliB, AliC and AliD. Since the authors have nine structures with five of them having clear electron density for the bound peptides. I think the most important question they should address is the peptide preference of each of the OBPs. They should define the peptide binding pocket for each amino acid as P1, P2, P3……, and discuss the amino acid preference of P1, P2, P3 ……

2. The authors performed mass spectrometry analysis of the oligopeptides captured by AliB and AmiA during purification. They observed distinct tendency to bind specific peptides. Given that they have the structures of AliB and AmiA, obviously they should map the distinct tendency back to the structures and provide a structural view of tendency.

Reviewer #2: (1) The authors must supply the validation reports for all the reported structures to verify deposition and to accurately judge the models. This manuscript cannot be accepted for publication without proper validation of the structures.

(2) The authors describe the domain-swapped dimer architecture of AliC, distinct from other OBPs. I think this point is important, however, the authors does not further elaborate or discuss it further. Whether this conformation is physiological or is caused by molecule packing during crystallization should be answered.

(3) The authors claim that the C-terminal �-helix (�19 in AliD) as the distinctive hallmark of all the Ali proteins, contributes to the stabilization of Lobe1 on the membrane (Line 231-233). The authors should supply proper biochemical assays, e.g. liposome co-sedimentation using OBP mutants lack of this �-helix.

(4) In the manuscript, although the authors report several complex structures of OBPs with oligopeptides and reveal the key residues involved in oligopeptide-binding structurally, it is still lack of proper biochemical verification. The authors should provide biochemical-binding assays (e.g. ITC or MST etc.) to confirm their structural observations.

(5) In the model of AF model, how AmiC or AmiD recognizes AliD in closed or open state. The authors should provide the critical sites involved in interaction of AliD with AmiC/AmiD and corresponding biochemical evidence to support their model. Whether the recognition of other OBPs by AmiC/AmiD is conserved? In addition, it is better that the panel A of Figure 5 should be shifted to the last panel.

(6) S6-10 Figs.: The authors show 2Fo-Fc maps with the peptide. I think this is a 2mFo-DFc? The authors need to also show the mFo-DFc maps of peptide as well. Additionally, a contour of 0.8 sigma is a bit low for displaying ligand density, especially given the high resolution, it is unclear if the ligand density is due to model bias. Was the ligand not well bound?

Reviewer #3: 1) The study lacks validation of the conclusions on substrate specificity. For example, the authors concluded that the first 3 residues of substrate peptides share common features that determine specificity for AliB. They also know how those features interact with the substrate-binding pocket. I think that this study would be strengthened if, based on their model, the authors generated a few protein variants predicted to be defective in substrate binding as well as a few artificial substrates that would be predicted to bind. Binding could be tested using standard biochemical binding assays (no structures required). Ideally, this could be done with a member of each the AliB-like and AmiA-like groups.

2) The authors identified peptides that were bound to purified Ami/Ai proteins that were produced in E. coli. Then, they determined the proteins that the peptides were derived from and discussed the various functions of these proteins (page 20). They noted that many of these proteins are ribosomal proteins or proteins associated with ribosomes. They also highlighted metabolic proteins and proteins involved in transcription, DNA replication or macromolecule transport. I do not understand the significance of analyzing the type of functions of these proteins (Fig. S11). The significance would have to be explained. To me, it is likely that there is an enrichment of ribosome-derived peptides because ribosomes are very abundant and large. Are most of the peptides derived from abundant proteins? It is yet to be determined if Ami proteins evolved to recognize peptides derived from abundant and conserved proteins such as ribosomal proteins. Moreover, the peptides that were trapped are derived from cytoplasmic proteins because the Ami/Ali proteins were produced in the cytoplasm of E. coli. Naturally, this is not the cellular compartment where Ami/Ali proteins function. Therefore, for the type of proteins (peptides) described by the authors to be relevant, they would have to be released from the cytoplasm likely through lysis. Given that S. pneumoniae undergoes autolysis, and if these peptides are also found in the cytoplasm of S. pneumoniae, is this transport system relevant in the context of the autolysis program of this bacterium? Currently, the Discussion section lacks an insightful discussion on the physiological role of this system. It might be worth considering this point if applicable.

**Part III – Minor Issues: Editorial and Data Presentation Modifications**

Reviewer #1: (No Response)

Reviewer #2: Note to authors: there are multiple errors of typo. Below is only an example for the authors’ reference.

(1) Line 411: the color of parentheses should be black.

(2) Line 547: the color of “AF” should be black.

(3) Line 740: the underline of unit “℃” should be removed.

(4) Figure 4D & S6-10 Figs.: Residue labeling should be consistent with others.

(5) Line 1159: “Figure S11.” should be “S11 Fig.”?

(6) Figure S11A: the X-axis labeling “Lenght” should be “Length” in x-axis.

Reviewer #3: 1) Line 179: "652); were..." should have a comma, not a semicolon.

2) Table S1: It would be better to use protein names as titles of columns instead of numbers.

3) In Materials and Methods: Please clarify that protein variants lacking their respective signal sequences were produced in the cytoplasm of E. coli.

PLOS authors have the option to publish the peer review history of their article (what does this mean?). If published, this will include your full peer review and any attached files.

Reviewer #1: No

Reviewer #2: No

Reviewer #3: No

Figure Files:

Data Requirements:

Reproducibility:

To enhance the reproducibility of your results, we recommend that you deposit your laboratory protocols in protocols.io, where a protocol can be assigned its own identifier (DOI) such that it can be cited independently in the future. Additionally, PLOS ONE offers an option to publish peer-reviewed clinical study protocols. Read more information on sharing protocols at https://plos.org/protocols?utm_medium=editorial-email&utm_source=authorletters&utm_campaign=protocols">https://plos.org/protocols?utm_medium=editorial-email&utm_source=authorletters&utm_campaig

---

## [Decision Letter · Decision Letter 1]

30 Apr 2024

Dear Prof. Hermoso,

We are pleased to inform you that your manuscript 'Molecular and Structural Basis of Oligopeptide Recognition by the Ami Transporter System in Pneumococci' has been provisionally accepted for publication in PLOS Pathogens.

Best regards,

Gongyi Zhang

Academic Editor

PLOS Pathogens

Michael Otto

Section Editor

PLOS Pathogens

Michael Malim

Editor-in-Chief

PLOS Pathogens

orcid.org/0000-0002-7699-2064

Reviewer Comments (if any, and for reference):

Reviewer's Responses to Questions

**Part I - Summary**

Reviewer #1: The authors have address my questions.

Reviewer #3: This manuscript presents a significant advance in our understanding of substrate recognition by the Ami ABC transporter system in Streptococcus pneumoniae. The work is of high quality. The revised manuscript has been improved in response to previous reviews. There is still more to understand about this important system, but this work is a solid piece of work that provides fundamental knowledge about this transporter. Its findings will impact our understanding of basic physiology of this bacterium (nutrient acquisition) and will guide future mechanistic studies on this transporter.

**Part II – Major Issues: Key Experiments Required for Acceptance**

Reviewer #1: (No Response)

Reviewer #3: (No Response)

**Part III – Minor Issues: Editorial and Data Presentation Modifications**

Reviewer #1: (No Response)

Reviewer #3: (No Response)

PLOS authors have the option to publish the peer review history of their article (what does this mean?). If published, this will include your full peer review and any attached files.

Reviewer #1: No

Reviewer #3: No

---

## [Editor Report · Acceptance letter]

29 May 2024

Dear Prof. Hermoso,

We are delighted to inform you that your manuscript, "Molecular and Structural Basis of Oligopeptide Recognition by the Ami Transporter System in Pneumococci," has been formally accepted for publication in PLOS Pathogens.

Best regards,

Michael Malim

Editor-in-Chief

PLOS Pathogens

orcid.org/0000-0002-7699-2064